

# Extensional exhumation of cratons: insights from the Early Cretaceous Rio Negro-Juruena belt (Amazonian Craton, Colombia)

Ana Fonseca[1], Simon Nachtergaele[1], Amed Bonilla[2], Stijn Dewaele[1], Johan De Grave[1]

[1]Laboratory for Mineralogy and Petrology, Department of Geology, Ghent University, Ghent, 9000, Belgium
5  [2]Servicio Geológico Colombiano, Dirección de Recursos Minerales, Bogotá, Colombia

*Correspondence to*: Ana Fonseca (anacarolina.LiberalFonseca@ugent.be)

**Abstract.** This study presents results from apatite fission track (AFT) thermochronology to investigate the thermal history and exhumation dynamics of the Rio Negro-Juruena basement, situated within the western Guiana Shield of the Amazonian Craton. AFT dating and associated thermal history modelling in South America has largely been restricted to the plate's margins (e.g. Andean active margin, Brazilian passive margin and others). Our paper reports on low-temperature 10 thermochronological data from the internal part of the western Guiana Shield for the first time. This area is part of a vast cratonic lithosphere that is generally thought to be stable and little influenced by Mesozoic and Cenozoic tectonics. Our data however show AFT central ages ranging from 79.1 ± 3.2 Ma to 177.1 ± 14.8 Ma with mean confined track lengths of ca. 12 µm. Contrary to what might be expected of stable cratonic shields, inverse thermal history modeling indicates a rapid 15 basement cooling event in the early Cretaceous. This cooling is interpreted as a significant exhumation event of the basement that was likely driven by the coeval extensional tectonics associated with back-arc rifts in the Llanos and Putumayo-Oriente-Maranon basins. The extensional tectonics facilitated both basement uplift and subsidence of the adjoining basins, increasing erosional dynamics and consequent exhumation of the basement rocks. The tectonic setting shifted in the late Cretaceous from extensional to contractional, resulting in reduced subsidence of the basins and consequential diminishing cooling rates 20 of the Guiana Shield basement. Throughout the Cenozoic, only gradual, slow subsidence occurred in the study area due to regional flexure linked to the Andean orogeny. Comparative analysis with low-temperature thermochronology data from other West Gondwana cratonic segments highlights that exhumation episodes are highly controlled by tectonic inheritance, lithospheric strength, and proximity to rift zones. This study underscores the complex interplay between tectonic events and the response of cratonic lithosphere over geological time scales and highlights extensional settings as an important 25 geological context for craton exhumation.

## 1 Introduction

Cratons, originating from the Greek term 'kratos' denoting 'strength,' are the vestiges of the Earth's initial lithosphere. While the term 'craton' is commonly applied to denote steady segments of Archean (>2.5 Ga) crust, its definition remains unbound by age connotations, since some of them might have only achieved their ultimate consolidation and stability in the



Proterozoic (Bleeker, 2003). Traditionally, they are described as regions that have achieved and maintained prolonged periods of tectonic and geomorphic stability, existing as relatively inert fragments of crust (e.g., Kusznir and Park, 1984; Artemieva, 2006; Manatschal et al., 2015; Salazar-Mora et al., 2018). The stability of cratons is thought to be partly influenced by their thicker lithosphere, often exhibiting a high effective elastic thickness that can exceed 100 km (Artemieva, 2006). Additionally, they are characterized by a relatively cold but compositionally buoyant upper mantle keel.

Recently, there has been increasing evidence challenging the concept of enduring craton stability and suggesting periodic disturbance within cratonic settings (e.g., Foley, 2008; Hu et al., 2018; Kusky et al., 2014; Liu et al., 2018; Snyder et al., 2017). Notably, this evidence includes periods of rifting and the formation of intracratonic aulacogens, occurrences of Mesozoic and Cenozoic magmatism, reports of intracratonic earthquakes, and even elevation changes on the craton's surface (e.g., Chimpliganond et al., 2010; Ault et al., 2013; Gordon et al., 2017; Ye et al., 2017; Rooney, 2020). In this context,

Bedle et al. (2021) present two plausible explanations for craton deformation and instability: either the craton's initial formation lacked the necessary thickness, buoyancy, or strength to maintain stability across dynamic conditions, or subsequent geological processes have altered its stability. Noteworthy among these destabilizing processes or conditions, as emphasized by Bedle et al. (2021), are shearing, proximity to buffer zones (e.g., orogens or mobile belts), interaction with mantle plumes, and rifting. A significant aid in unravelling the chronological sequence of these events comes from low-

temperature thermochronology, which effectively constrains the episodes of cooling and heating within the Earth's upper crust (see Kohn and Gleadow, 2019 for a review). These episodes of crustal cooling and heating are closely linked to geological phenomena such as kilometer-scale erosion, sedimentary burial, faulting, and magma emplacement. As a result, low-temperature thermochronology play a pivotal role in providing insights into deciphering the intricate history of potential geodynamic processes influencing the (in)stability of cratons.

The Amazonian Craton is a vast craton exposed in the Amazonian rainforest in South America and was once part of the West Africa-Amazonian Craton (Fig. 1). It can be subdivided into the Guiana Shield to the north of the Amazon River Basin and the Guaporé Shield to the south (Fig. 1). The West Africa - Amazonian Craton underwent amalgamation with other cratonic landmasses along several orogenic belts during the Brasiliano-Pan-African orogeny (ca. 600 Ma). This event culminated in the emergence of the West Gondwana paleocontinent (Buiter and Torsvik, 2014). Subsequently, the West

Africa - Amazonian Craton experienced disruption through Jurassic to early Cretaceous rifting in the equatorial Atlantic region. The effects of different periods of Mesozoic extensional tectonics associated with the break-up of the Pangea and Gondwana supercontinents allowed basin formation along the proto-Caribbean margin and into the intracontinental cratonic regions of South America, (Fig. 1; Vaz et al., 2007). This rift event ultimately led to the partitioning of the Amazonian and West Africa cratonic counterparts, positioning them in South America and Africa, respectively.





**Figure 1: a) Digital elevation map (SRTM, USGS) of northern South America with main tectonic structures, terranes and basins indicated (modified from Gómez et al., 2019). Our study area and the regions where previous low-temperature thermochronology date were obtained are delineated by the dashed boxes. b) E-W schematic regional structural section across the central region of the Northern Andes into the western Amazonian Craton (Modified from Cediel et al. 2003).**

One section among the various different geochronological domains or provinces constituting the Amazonian Craton is the Rio Negro-Juruena belt, located at the western margin of the craton (Fig. 1). The Rio Negro-Juruena belt is a region with a controversial history, in Colombia, consisting of Paleoproterozoic (1.80-1.50 Ga) gneisses in amphibolite facies and



granitoids, and Mesoproterozoic (1.40-1.34 Ga) anorogenic magmatism (Teixeira et al., 1989; Tassinari and Macambira, 1999; Cordani et al., 2016; Bonilla et al., 2021, 2023). During the Jurassic to the early Cretaceous, the belt delimited the
Llanos and Putumayo-Oriente-Maranon basins that were evolving as a back-arc rifting zone (Fig. 1; Horton, 2018; Cediel and Shaw, 2019; Guerrero et al., 2020). The Rio Negro-Juruena belt has been investigated with several geochronometers such as zircon U-Pb, whole rock Sm-Nd, whole-rock Rb-Sr and mica K-Ar (Bonilla et al., 2021, 2023; Cordani et al., 2016; Santos et al., 2000; Tassinari and Macambira, 1999), but only one study involved low-temperature techniques as apatite fission track thermochronology (Bonilla et al., 2020). Nevertheless, the application of these techniques can enrich our
understanding of the more recent evolution of craton evolution and (in)stability (Kohn and Gleadow, 2019). Furthermore, the gathered thermochronological data has the potential to yield significant tectonic insights on the evolution of craton-adjacent sedimentary basins (e.g. Guerrero et al., 2020; Horton et al., 2010; Hurtado et al., 2018; Vallejo et al., 2017). Up to the present, low temperature thermochronology studies in northern South America were mainly directed towards the Andean orogen (e.g. Amaya et al., 2017; Parra et al., 2009; Pérez-Consuegra et al., 2021; Spikings et al., 2000; Villagómez et al.,
2011; Villagómez and Spikings, 2013). This study specifically aims at filling part of this gap and reconstructing the thermal history of the large and under-explored cratonic Guiana Shield, with implications for gaining deeper insights into the evolution of some of the most prominent Colombian sedimentary basins. We analysed 20 crystalline basement samples with apatite fission track (AFT) thermochronology in order to reconstruct the thermal history of the basement rocks from the Rio Negro-Juruena belt of the western Guiana Shield. Additionally, we compare our findings with prior thermochronological
research of cratonic areas that once composed West Gondwana to gain insights into the mechanisms underlying Phanerozoic craton destabilizing processes.

## 2 Geological Setting

During the Proterozoic, the Amazonian–West Africa Craton further amalgamated with the development of accretionary orogenic belts along the western margin of its Archean nucleus (Tassinari and Macambira, 1999; Santos et al., 2006). These
belts are referred to as the Ventuari-Tapajós (ca. 2.0 - 1.8 Ga), Rio Negro-Juruena (ca. 1.8 – 1.5 Ga), Rondonian-San Ignacio (ca. 1.5-1.3 Ga), and Putumayo (1.45–0.98 Ga) belts. Their formation ages decrease towards the current Andean margin of South America (Fig. 1; Ibanez-Mejia et al., 2011; Cordani and Teixeira, 2007; Tassinari and Macambira, 1999). Our study area comprises the NW sector of the Amazonian Craton, in the Rio Negro-Juruena belt, near the projected suture with the Ventuari–Tapajós belt (Fig.1), as traced by Cordani and Teixeira (2007). The area includes the boundary between Colombia
with Venezuela and Brazil (Fig. 3), where the geology has remained relatively unexplored due to the challenges posed by limited accessibility, the absence of road infrastructure, and extensive vegetation cover. In the western and north direction, towards the Andes and Venezuela, and in the southward direction, towards the Amazon River, the Precambrian basement is buried under a thick sedimentary cover of the Llanos and Putumayo-Oriente-Maranon basins, which has a record at least since the Paleozoic (Fig. 1) (Moreno-López and Escalona, 2015). Here, we provide a concise overview of the tectonic



evolution of the research area. For more complete information, we recommend, amongst others, the publications by Cediel and Shaw (2019) and Bonilla et al. (2021) and references therein.

## 2.1 Precambrian

The basement rocks of the study area encompass Paleoproterozoic to Mesoproterozoic (1.85 – 1.50 Ga) (meta)granitoid rocks, mostly calc-alkaline gneisses, granites, and migmatites that experienced amphibolite facies metamorphism (Fig.3;

Bonilla et al., 2021; Cordani et al., 2016). Based on available Rb-Sr and Sm-Nd data, these rocks are thought to have originated from a sequence of Proterozoic subduction events involving juvenile magmatic arcs (Tassinari and Macambira, 1999; Cordani et al., 2016). Some areas contain low-grade metamorphosed volcanic-sedimentary sequences and intraplate Mesoproterozoic anorogenic granites (Fig. 3). K-Ar dating on micas of these rocks indicates a widespread intraplate heating event, with basement temperatures exceeding 300°C around 1.2 – 1.3 Ga (Cordani et al., 2016). The thermal event was

constrained by U-Pb apatite ages and it has been related to large anorogenic magmatism in a continental rift (Bonilla et al., 2023)

The Amazonian-West Africa Craton became part of the Rodinia supercontinent around 1.0 Ga ago through a continental collision of its western margin with southern Laurentia (Kroonenberg, 1982; Ibanez-Mejia et al., 2011; Li et al., 2008), while during the Neoproterozoic (ca. 700-600 Ma), break-up of Rodinia separated them again (Li et al., 2008; Cawood and

Pisarevsky, 2006). In the late Neoproterozoic (ca. 600-500 Ma), the margins of the Amazonian-West Africa craton collided with the eastern South American and African cratons, along suturing orogenic belts, eventually resulting in the formation of West Gondwana (Johansson, 2014; Li et al., 2008).

## 2.2 Phanerozoic

For the entire Paleozoic, more specifically from the late Neoproterozoic to the early Mesozoic, the Amazonian-West Africa

craton was part of the continental interior of Gondwana and Pangea. It remained at the hinterland of the palaeocontinents until their break-up. Early Mesozoic rifting affected both sides of the study region. On the western margin of northwestern Gondwana, arc magmatism and marine incursions occurred during a prevailing extensional regime (Spikings et al., 2019; Cardona et al., 2010). The current Andean orogenic cycle started during the Jurassic, following a pause in subduction between the late Permian and late Triassic (Ramos and Aleman, 2000). Since then, oceanic subduction has been continuous

(Pankhurst et al., 2000; Coira et al., 1982; James, 1971). During the Jurassic and Cretaceous, the northern Andean margin was characterized by the development of a prominent volcanic arc with back-arc extension (Spikings et al., 2015; Maloney et al., 2013; Horton et al., 2010). The Putumayo-Oriente-Maranon Province and Llanos basins (Fig. 1) experienced rifting and tectonic subsidence due to this back-arc extension (Parra et al., 2009; Horton et al., 2010), accumulating primarily Cretaceous shales and marls within a shallow marine environment (Fig. 2; Jiménez et al., 2020; Villamil and Arango, 1998;

Cooper et al., 1995). Occasionally, sandstone and limestone units prograde locally. In general, during the Aptian-Albian (ca. 120 – 100 Ma), the deposits accumulated in continental and shallow-marine to tidal environments, while the Cenomanian-



Coniacian (ca. 100 – 85 Ma) strata were predominantly formed in a deeper marine, shelf environment (Fig. 2; Villamil and Arango, 1998; Cooper et al., 1995). Albian magmatism (ca. 103 Ma) is recorded in the Colombian basement in the continental interior (Ibañez–Mejia et al., 2014). A Santonian-Maastrichtian (ca. 85 – 65 Ma) regressive sequence comprises

deposits from both marine and coastal plain settings (Fig. 2). Marine deposition was abruptly terminated during the early Maastrichtian (ca. 70 Ma) due to the accretion of the Western Cordillera that also uplifted the basin area in the hinterland (Fig. 2; Cooper et al., 1995). From the early Cenozoic to the present (ca. 70 – 0 Ma), the Putumayo and Llanos basins acted as foreland basins, subsiding due to the orogenic load of the Andean terrains.

**Figure 2: Simplified stratigraphic chart of the Llanos and Oriente basins providing the stratigraphic context for published detrital zircon U-Pb data that we depict in Fig. 5 (modified from Mora et al. 2006; Parra et al., 2009; Guitiérrez et al., 2019; Vallejo et al., 2021). The ages in blue refer to the time range of basement cooling constrained by our AFT data.**



Meanwhile, in the equatorial Atlantic passive margin between northern South America and western Africa (Sapin et al., 2016; Deckart and Gilbert, 1997) incipient rifting is thought to have started in the early Jurassic, coinciding with significant

volcanic activity documented in the Central Atlantic Magmatic Province ca. 200 Ma ago (CAMP; Marzoli et al., 2018; Marzoli et al., 1999). In the Cretaceous, the extension ultimately transpired through transpressional and transtensional deformation until the complete rupture of the paleocontinent in the Aptian (ca. 120 Ma) and consequential opening of the Atlantic Ocean. This rupture resulted in the effective separation of the Amazonian and West Africa cratons, which became situated within South America and Africa, respectively.

In the interior of the Amazonian craton, the Takutu rift basin (Fig. 1) evolved as an aulacogen related to the opening of the equatorial Atlantic Ocean (Vaz et al., 2007). Triassic tholeiitic dykes from the CAMP and early Cretaceous flood basalts with ages of ca. 140 Ma intruded the Takutu rift basement (Reis et al., 2006). Moreover, alkaline nepheline-bearing syenites of $111 \pm 1$ Ma and $116 \pm 3$ Ma are found in the surrounding basement adjacent to the basin (Figueredo et al., 2018). Sedimentation in the Takutu rift basin occurred between the early Jurassic and the early Cretaceous (Sapin et al., 2016). The

deposits have an estimated thickness of around 7300 meters, encompassing continental volcanic-sedimentary sequences (Vaz et al., 2007).

## 3 Samples and methods

Samples were taken from outcrops of the Rio Negro - Juruena belt along one NNW-SSE, and three ENE-WSW oriented profiles along the Guainía-Negro, Inírida and Cuiari rivers (Table 1; Fig. 1 and 3). Previously obtained zircon and apatite U-

Pb (ZUPb-APUPb) ages for these samples indicate that they were all formed between 1.85 and 1.50 Ga, except for sample 502, which had a slightly younger weighted mean ZUPb age of 1.40 Ga (Bonilla et al., 2021). A regional Mesoproterozoic thermal event results in the reseating of APUPb ~ 1.40 Ga (Bonilla et al., 2023).

The apatite fission-track (AFT) method was used to reconstruct the thermo-tectonic evolution of the belt. The AFT method is based on the natural fission decay of 238U which creates lattice damage trails, known as fission tracks, within the apatite

crystal. Over time, these fission tracks accumulate at a constant rate, and the number of tracks per unit area serves as a measure of the AFT age. The AFT age represents a cooling age since fission tracks in apatite become stable and are retained at temperatures below ca. 120°C. This temperature corresponds to a depth of about 4 km in the continental crust, when considering a geothermal gradient of 25-30°C/km. Between approximately 120°C and 60°C, fission tracks in apatite are preserved but are shortened due to the thermal restoration of the crystal lattice. This temperature range is known as the

apatite partial annealing zone (APAZ; Wagner and Van den haute, 1992 and references therein). At temperatures higher than ca.120°C, fission tracks in apatite are annealed completely over geological time scales (e.g., Gleadow and Brown, 2000). As a result, the AFT length provides information about the past temperatures experienced by the apatite-bearing rock during its evolution in the Earth's crust (Gleadow et al., 1986; Ketcham et al., 2007). The annealing behavior and shortening of apatite



fission tracks can hence be exploited for thermal history modeling (e.g., Gallagher, 2012; Ketcham, 2005; Laslett et al., 1987).

| Sample | Latitude (°N) | Longitude (°W) | Lithology | Elevation (m) |
|:------:|:-------------:|:--------------:|:---------:|:-------------:|
| 124 | 1.778028 | 68.917570 | Porphyritic syenogranite | 154 |
| 134 | 1.959167 | 68.364610 | Monzogranite | 135 |
| 135 | 1.834278 | 68.707450 | Quartz monzodiorite | 147 |
| 164 | 2.718608 | 67.948862 | Monzogranite | 98 |
| 168 | 2.798120 | 67.834094 | Monzogranite | 97 |
| 170 | 2.827199 | 67.799111 | Monzogranite | 111 |
| 171 | 2.728289 | 67.567755 | Migmatite | 88 |
| 175 | 2.589041 | 67.420448 | Leucosome | 84 |
| 176 | 2.529057 | 67.365048 | Migmatite | 90 |
| 180 | 2.166866 | 67.181351 | Migmatite | 86 |
| 182 | 1.914885 | 67.066977 | Monzogranite | 81 |
| 501 | 2.792967 | 69.281958 | Syenogranite | 133 |
| 502 | 2.893865 | 69.061838 | Monzogranite | 128 |
| 505 | 2.967441 | 68.714423 | Quartzsyenite | 112 |
| 506 | 2.970117 | 68.667985 | Quartzsyenite | 107 |
| 510 | 3.056191 | 68.506132 | Monzogranite | 107 |
| 512 | 3.208222 | 68.218030 | Monzogranite | 95 |
| 515 | 3.407566 | 67.973478 | Monzogranite | 86 |
| 516 | 3.458146 | 67.981013 | Monzogranite | 93 |

**Table 1: Sample locations of the analysed samples in this study. Lithology and elevation above sea level is also indicated.**

In the present work, the AFT analyses were conducted using the external detector (ED) method (Fleischer et al., 1975). All samples were irradiated using one irradiation container in the well-thermalized channel X26 of the BR1 reactor (Belgian Centre for Nuclear Research, SCK, Mol, Belgium). Thermal neutron fluence was monitored using four ED-covered Uranium-doped glass dosimeters IRMM-540R (De Corte et al., 1998), spatially distributed in the irradiation package. Latent fission tracks in apatite were etched using a 5.5M nitric acid solution for 20s at 21°C. Induced fission tracks were revealed in the ED muscovite with 40vol% HF etchant for 40min at 21°C. The Overall Weighted Mean Zeta (OMWZ) based on both Durango (McDowell et al., 2005) and Fish Canyon Tuff (Hurford and Hammerschmidt, 1985) apatite age standards for analyst SN using IRMM-540R glass dosimeters is $286.3 \pm 3.9$ ($1\sigma$) a*cm². The spontaneous and induces fission tracks in both the standards and samples were manually counted using a Nikon Eclipse Ni-E microscope. The microscope was equipped with a DS-Ri2 camera and set to a magnification of 1000x. To facilitate the counting process, the Trackflow software was employed (Van Ranst et al., 2020). The lengths of confined fission tracks and their angles to the c-axis were measured using the same microscope setup. The sub-horizontal confined tracks were corrected for inclination.





**Figure 3: Geological map of the study area with sample site locations and apatite fission track (AFT) results (modified from Almeida and Mendes, 2021).**



## 4 Results

The AFT results on twenty apatite samples from the Rio Negro-Juruena belt are shown in Table 2 and geographically displayed in Fig. 3. AFT central ages range from $79.1 \pm 3.2$ Ma to $177.1 \pm 14.8$ Ma and 14 out of 20 samples yield central
ages between 95 Ma and 120 Ma. Where possible, 20 or more apatite grains were analysed per individual sample. Three samples (168,192 and 506) fail the chi-squared test on 95% confidence level (i.e. $P(\chi^2) < 0.05$). The study area is characterized by large flood plains and rare inselbergs from which most samples were collected, thus only a limited elevation difference (<100m) exists between all samples. Therefore, no age-elevation scatter plots were made. A minimum of 50 confined tracks could be measured for 12 out of 20 samples, and 100 for 9 samples. These samples present uncorrected mean
track lengths between 11.6 and 12.5 µm, standard deviations from 1.4 µm to 2.0 µm, and a negatively skewed distribution that is typical from samples with some degree of track annealing.

## 5 Thermal History Modelling

Samples with more than 50 confined track length measurements underwent inverse thermal history modeling using QTQt 5.6.3 (Gallagher, 2012). QTQt uses the transdimensional Markov Chain Monte Carlo search method to find a range of
probable time-temperature path solutions (Gallagher, 2012) based on the sample's AFT dataset to reconstruct its thermal history. We applied 105 burn-in and post burn-in iterations. The present-day surface temperature constraint was set as $30 \pm 5$ °C (CDA, 2020) for all models. The Rio Negro Juruena basement is an area with dense vegetation and few outcrops. There is also limited accessibility for structural and geological observations and absence of Mesozoic sediments. Hence, it was only possible to define reliable constraints to the thermal history models based on the other available geochronology results. For
some of the model runs we chose a Mesoproterozoic constraint ($1250 \pm 50$ Ma; $350 \pm 50$°C) based on the K-Ar and U-Pb apatite dating of the metamorphosed volcanic-sedimentary sequences of the study area (Cordani et al., 2016; Bonilla et al., 2023). Constraints at higher temperatures and further back in time (e.g. the available ZUPb data) would not affect the modeling (Abbey et al., 2023). Details concerning the thermal history modeling and the strategies used are given in Supplement S4.
Initially (modeling strategy M1), we modeled each sample employing the default priors in QTQt (Gallagher, 2012). These priors include a temperature range of $70 \pm 70$°C and an age range of central age ± central age. Track length data were converted to c-axis projected lengths (Ketcham et al., 2007a). The multi-kinetic apatite fission track annealing model of Ketcham et al. (2007b) with the etch pit diameter (Dpar) as kinetic parameter was used. The results for the expected models (Supplement S4) indicate a common thermal history for the entire study area. They show a gradual slow cooling through the
APAZ (at rates of ca. 0.4°C/Ma) from the Triassic to the late Cretaceous.

One representative sample (sample 171) was chosen to test the variability of these thermal histories in five other modelling strategies (i.e., M2 to M6; Supplement S4). Modeling strategy M2 was performed using the c-axis projection for the track length data and default priors in QTQt. For modeling strategy M3, the priors were widened to $300 \pm 300$ Ma for time and




| Sample | n | ρs (10⁶tr/cm²) | Ns | ρi (10⁶tr/cm²) | Ni | ρd (10⁶tr/cm²) | Nd | ρs/ρi | Pooled age ± 1σ (Ma) | Central age ± 1σ (Ma) | P(χ²) | lm | σ | lmc | σc | nl | Dpar |
|---|---|---|---|---|---|---|---|---|---|---|---|---|---|---|---|---|---|
| **124** | **25** | **1.050 (0.033)** | **984** | **0.733 (0.028)** | **702** | **0.581 (0.008)** | **5290** | **1.544** | **115.5 ± 5.9** | **116.3 ± 6.7** | **0.22** | **12.0** | **1.6** | **13.6** | **1.1** | **71** | **1.3** |
| 134 | 20 | 0.762 (0.030) | 653 | 0.342 (0.020) | 300 | 0.581 (0.008) | 5623 | 2.319 | 175.5 ± 12.7 | 177.1 ± 14.8 | 0.19 | - | - | - | - | - | 1.3 |
| 135 | 20 | 0.729 (0.024) | 909 | 0.430 (0.018) | 551 | 0.581 (0.008) | 5159 | 1.816 | 135.8 ± 7.6 | 136.4 ± 9.3 | 0.10 | - | - | - | - | - | 1.4 |
| 164 | 20 | 0.394 (0.019) | 453 | 0.283 (0.016) | 320 | 0.553 (0.007) | 5624 | 1.433 | 111.2 ± 8.3 | 111.2 ± 8.4 | 0.96 | - | - | - | - | - | 1.2 |
| 168 | 28 | 0.296 (0.016) | 329 | 0.242 (0.007) | 238 | 0.557 (0.007) | 5688 | 1.569 | 109.2 ± 9.4 | 103.4 ± 12.0 | 0.02 | - | - | - | - | - | 1.3 |
| **170** | **25** | **0.554 (0.016)** | **1147** | **0.324 (0.012)** | **673** | **0.558 (0.007)** | **5714** | **1.856** | **134.8 ± 6.8** | **135.5 ± 7.6** | **0.22** | **12.0** | **1.5** | **13.5** | **1.1** | **65** | **1.2** |
| **171** | **25** | **1.040 (0.031)** | **1101** | **0.725 (0.026)** | **778** | **0.560 (0.007)** | **5746** | **1.420** | **112.5 ± 5.5** | **112.1 ± 6.1** | **0.20** | **11.6** | **2.0** | **13.4** | **1.2** | **100** | **1.3** |
| **175** | **25** | **0.754 (0.020)** | **1060** | **0.575 (0.020)** | **803** | **0.562 (0.007)** | **5771** | **1.390** | **105.4 ± 5.1** | **105.4 ± 5.3** | **0.73** | **11.9** | **1.7** | **13.3** | **1.5** | **100** | **1.1** |
| **176** | **20** | **3.050 (0.077)** | **1570** | **3.110 (0.078)** | **1591** | **0.564 (0.007)** | **5789** | **0.980** | **79.1 ± 3.0** | **79.1 ± 3.2** | **0.85** | **11.9** | **1.7** | **13.5** | **1.2** | **100** | **1.3** |
| **180** | **25** | **0.639 (0.019)** | **1101** | **0.581 (0.018)** | **1023** | **0.572 (0.007)** | **5830** | **1.080** | **87.5 ± 4.0** | **86.6 ± 4.7** | **0.10** | **11.7** | **1.7** | **13.4** | **1.0** | **100** | **1.5** |
| **182** | **30** | **0.883 (0.023)** | **1462** | **0.746 (0.021)** | **1256** | **0.573 (0.008)** | **5824** | **1.290** | **94.8 ± 3.9** | **96.1 ± 4.8** | **0.07** | **12.5** | **1.6** | **13.9** | **1.0** | **100** | **1.4** |
| 192 | 10 | 4.320 (0.129) | 1114 | 3.640 (0.120) | 919 | 0.581 (0.008) | 5623 | 1.176 | 100.0 ± 4.7 | 97.0 ± 7.3 | 0.00 | - | - | - | - | - | 1.5 |
| 501 | 25 | 0.359 (0.018) | 405 | 0.197 (0.013) | 225 | 0.574 (0.008) | 5811 | 1.923 | 146.3 ± 12.3 | 146.3 ± 12.5 | 0.78 | - | - | - | - | - | 1.3 |
| **502** | **25** | **1.630 (0.044)** | **1397** | **1.140 (0.036)** | **985** | **0.575 (0.008)** | **5798** | **1.489** | **115.7 ± 5.1** | **116.8 ± 6.3** | **0.14** | **11.8** | **1.6** | **13.5** | **1.0** | **100** | **1.3** |
| **505** | **20** | **1.720 (0.042)** | **1692** | **1.270 (0.036)** | **1221** | **0.576 (0.008)** | **5783** | **1.437** | **113.3 ± 4.5** | **113.3 ± 4.8** | **0.56** | **11.7** | **1.4** | **13.4** | **1.1** | **100** | **1.4** |
| **506** | **25** | **2.180 (0.047)** | **2193** | **1.670 (0.041)** | **1634** | **0.577 (0.008)** | **5763** | **1.409** | **109.9 ± 3.9** | **110.9 ± 5.6** | **0.00** | **12.2** | **1.6** | **13.7** | **1.1** | **100** | **1.3** |
| **510** | **20** | **1.490 (0.045)** | **1092** | **1.090 (0.038)** | **804** | **0.578 (0.008)** | **5735** | **1.452** | **111.3 ± 5.4** | **111.5 ± 6.2** | **0.24** | **12.3** | **1.5** | **13.8** | **1.2** | **100** | **1.3** |
| 512 | 25 | 0.307 (0.016) | 380 | 0.235 (0.014) | 288 | 0.580 (0.008) | 5509 | 1.400 | 108.7 ± 8.6 | 108.7 ± 8.8 | 0.60 | - | - | - | - | - | 1.3 |
| **515** | **25** | **0.858 (0.025)** | **1184** | **0.677 (0.022)** | **922** | **0.581 (0.008)** | **5462** | **1.302** | **105.9 ± 4.9** | **105.8 ± 5.4** | **0.22** | **12.4** | **1.8** | **13.8** | **1.2** | **85** | **1.3** |
| 516 | 18 | 0.348 (0.022) | 254 | 0.250 (0.018) | 183 | 0.581 (0.008) | 5404 | 1.418 | 114.4 ± 11.2 | 113.1 ± 12.2 | 0.44 | - | - | - | - | - | 1.1 |

**Table 2: Apatite fission track data of the analysed samples. The number of analysed grains (n), spontaneous (ρs) and induced track density (ρi) and induced track density in the glass dosimeter (ρd) with their 1σ uncertainties are given. The number of spontaneous tracks (Ns), induced tracks (Ni) and interpolated value for the glass dosimeter (Nd) are also displayed. The pooled age and central age are expressed in Ma. The p-value for the chi-squared test is also given. The raw mean track length (lm) and c-axis projected (lmc) mean track length, standard deviation (σ: raw; σc: c-axis projected) and number of measured tracks (nl) from confined track length measurements are indicated. In bold are sample data that were modelled with QTQt.**

150 ± 150°C for temperature. The results obtained from M2 and M3 closely resembled those obtained from modelling strategy M1 in which the expected t,T-path displays a continuous and slow cooling through the APAZ during the Meso-Cenozoic. Acceptance rates for time and temperature and birth and death were however not satisfactory for these strategies (see Supplement S4).

Modelling strategies M4 and M5 assumed an extra constraint for each model. An artificial zircon fission track age (closure temperature ca. 200°C) equals 200 ± 20 Ma was added for M4 model in order to test the hypotheses of the basement being at temperatures hotter than the APAZ in the early Jurassic. As an opposite hypothesis, an artificial exposure close to surface temperatures in 200 ± 20 Ma was added for M5 model, in this case, the basement would be colder than the APAZ temperatures in the early Jurassic. Interestingly, the expected models, which exhibited a consistent slow cooling trend in the M1, M2, and M3 modeling strategies, showed rapid cooling during the early Cretaceous in both M4 and M5 models. Acceptance rates for time and temperature and birth and death were in addition satisfactory.

In modeling strategy M6 we did not use c-axis projected confined track lengths, nor additional constrains. C-axis projection is a commonly employed technique to correct the AFT data for track orientation. It is based on the assumption that confined



tracks tend to be longer in directions parallel to the crystallographic c-axis and shorter perpendicular to the c-axis (Donelick,

1991, 1999; Ketcham et al., 2007b). The AFT community has extensively embraced this correction as the standard length correction. However, our length data deviates from the assumption of angle-length correlation showing no apparent anisotropy in the length distribution (Supplement S3). In this case, we decided to also test models without c-axis projection, such as in our M6 modelling protocol, that could be more reliable considering our more isotropic data. The M6 expected model also displays an early Cretaceous rapid cooling through the APAZ. Acceptance rates for time and temperature and

birth and death were satisfactory.

Finally, all samples were modeled together in batch modeling strategies M7 and M8. In M7, a c-axis projection was implemented, whereas in M8 not. The results for M7 and M8 were similar. The expected models displayed a fast cooling (at rates of ca. 50°C/Ma) in the early Cretaceous and residence close to the upper APAZ (ca. 60 °C) by the end of late Cretaceous (ca. 65 Ma) followed by slow cooling to surface temperatures. Acceptance rates for time and temperature and

birth and death were satisfactory.

## 6 Discussion

### 6.1 Inverse thermal history modeling

The inverse thermal history models (Supplement S4) reveal that the investigated rocks in the Rio Negro-Juruena basement experienced a common cooling history within the APAZ temperature range (ca.120 °C to 60 °C), with no significant

differences between samples. Despite the presence of intersecting lineaments suggesting potential faults (Fig. 1 and 3), Phanerozoic fault activity does not seem to have caused substantial differential displacements within the basement samples and this part of the Guiana Shield more or less acted as a single block in response to the Phanerozoic fault activity. Reactivation of faults can induce differential movements and uplift and subsequent erosion, resulting in variations in the rates of cooling. The similarity in paleo crustal depth levels of the exposed Precambrian rocks in the area (Bonilla et al.,

2023), along with the low seismic activity in the region (Pérez-Gussinyé et al., 2009), further supports limited differential younger reactivation of faults (Cenozoic). Consequently, the homogeneous rock cooling history observed in the study area aligns well with independent geological evidence, which indicates limited inter-sample reactivation.

The AFT data fits well with two main thermal history model solutions. The first solution is acquired in the M1, M2, and M3 modelling strategy (Supplement S4) and consists of a gradual and continuous cooling process through the APAZ during the

Cretaceous, with similar rates persisting until surface temperature conditions are reached as the rocks exhumed. The second solution is acquired in the M4, M5, M6, M7, and M8 modelling approaches and implies a rapid cooling event in the early Cretaceous, specifically between the Berriasian and Aptian (140 – 110 Ma ago), that caused the samples to reach the upper APAZ (ca. 65 °C). This event is followed by a gradual slow cooling to reach surface temperatures. The second solution is supported by four main additional arguments as the most plausible for our data.





The primary argument revolves around the fact that our study area exhibits a common cooling history, which justifies the batch of individual sample data into a one larger dataset as an effective approach for generating more reliable models. Modeling strategies M7 and M8 adopt this approach. The resulting models indicate rapid cooling through the APAZ during the early Cretaceous, representing the second solution. These models have been demonstrated to be well-resolved, showing consistency between the predicted and observed AFT data parameters (see Supplement S4). Therefore, it appears that M7 and M8 offer reasonable and plausible solutions, supporting the second solution as the preferred one.

The second argument focuses on analysing the maximum likelihood (MaxLike) paths in the M1 models. The MaxLike model represents the time-temperature path that best fits the data. Although in some circumstances it may exhibit a complex and geologically unrealistic solution (Abbey et al., 2023), to our specific dataset, these models generally show straightforward and realistic time-temperature paths. For almost all samples, the MaxLike paths indicated a rapid cooling event from the late Jurassic to the early Cretaceous (see Supplement S4). Thus, considering the geologically plausibility of the results and the consistent patterns observed across different samples, it is reasonable to conclude that the MaxLike models, which align with the second solution, provide a credible approximation of the true cooling history.

The third argument stems from the analysis of the modeling results when additional constraints are applied. By incorporating different time-temperature constraints in the modeling strategies M4 and M5, a similar accelerated cooling trend during the Cretaceous is observed for both strategies (Supplement S4). This indicates that the presented cooling event is not significantly affected by the inclusion of potentially new thermochronological or geological information from the past thermal history. In other terms, regardless of whether the dated rocks were deep in the crust or near the surface in the Jurassic, Cretaceous cooling, i.e., the second solution, is required to align with our data.

Finally, applying c-axis correction seems to be inadequate for the data, as discussed in section 5. Consequently, modeling strategies that exclude this correction tend to avoid overcorrections. Both the M6 and M8 models, which don't include the c-axis projection, show a rapid cooling trend in the early Cretaceous, in line with the second solution we discussed. This indicates that the M6 and M8 models support to considering the second solution as the preferred choice.

In conclusion, our AFT data provides strong evidence for a significant rapid cooling event during the early Cretaceous in the Rio Negro-Juruena belt. Out of all the models that favoured this solution, Model M8 is selected for subsequent discussions (Fig. 4).

## 6.2 Early Cretaceous cooling (140 – 110 Ma)

The early Cretaceous cooling event constrained by our data is most likely a result of the exhumation of the Rio Negro-Juruena belt due to regional denudation. We ruled out the thermal effects of the CAMP plume as the cause of the observed cooling event due to two compelling reasons. Firstly, the plume's influence in the crust is estimated to have ended ca. 200 Ma ago, which predates the cooling event constrained by our models (Fig. 4). Secondly, when examining previous data from regions impacted by mantle plumes, there is little to no evidence of the influence of plume impingement in the AFT system, even at relatively close distances (e.g., Sahu et al., 2013; Fonseca et al., 2020), which indicates that the crustal heating



resulted from the plume probably is very limited in the upper crust. Therefore, the early Cretaceous cooling event is most likely attributed to the erosion of the upper crust with the subsequent deposition of the produced clastic sediments in adjacent

basins, leading to the exhumation of the craton as it appears today.

**Figure 4: a) Thermal history model derived from the modelling strategy M8 (see Supplement S4), which was chosen as the representative for the study area (section 6.1). b) Schematic map of the tectonic compartments of the Llanos Basin during the Cretaceous and their tectonic subsidence in meters (from Sarmiento-Rojas et al., 2006). c) The tectonic subsidence of the**

**Cundinamarca (A), Cocuy (B), and Llanos Orientales (C) compartments of the basin (from Sarmiento-Rojas et al., 2006). d) Block diagram illustrating the configuration of the Llanos Basin in the early Cretaceous (modified from Cooper et al., 1995). e) Main geodynamic events affecting northern South America in the Mesozoic and Cenozoic. CAMP: Central Atlantic Magmatic Province.**

In the early Cretaceous, the northwestern margin of South America was evolving in a back-arc extensional setting, marked by the formation of rifts (Fig. 4; León et al., 2019; Zapata et al., 2019; Cardona et al., 2010). These rifts created depocenters

in the Llanos Basin and the Putumayo-Oriente-Maranon sedimentary Province (Fig. 1), where early Cretaceous marine sediments started to accumulated over the Jurassic volcanic and siliciclastic units (Fig. 2; Kammer and Sánchez, 2006;



Sarmiento-Rojas et al., 2006). Provenance analysis, including petrography and detrital zircon geochronology from various localities (Guitiérrez et al., 2019; Vallejo et al., 2019; Guerrero et al., 2020), suggest that before the early Cretaceous these basins were mainly sourced by Paleozoic igneous and metamorphic rocks from the Andean basement, located at the western

margin of the basins (Fig. 5; Horton et al., 2010; Hurtado et al., 2018; Cardona et al., 2010). Detrital zircon U-Pb analysis, however, reveals that Paleozoic sources from the Andean basement disappear, show a systematic decrease in Grenville-aged basement detritus, and a corresponding increase in Paleoproterozoic basement signatures that are only found in the eastern parts of the Guyana shield, including the Rio Negro-Juruena basement (Fig. 5; Horton et al., 2010). The provenance analysis confirms that the Rio Negro-Juruena belt experienced erosion during the early Cretaceous and the detrital sediments

resulting from the erosion were deposited in the aforementioned depocenters. Moreover, the change in the source area that occurred at the Jurassic-Cretaceous boundary suggests tectonic reorganization of the area during that time.

Our interpretation suggests that during the Jurassic, our study area could be characterized by a low-elevation plain, resulting in exceptionally low erosion rates and limited sediment production from the Rio Negro-Juruena basement to adjacent basins (Horton, 2018). This idea is supported by the existence of pre-cretaceous basemen-highs and sedimentary basins to the north

and west of the study area (e.g. Arauca Graben, Chiribiquéte sub-basin). This suggests that part of the basement in our study area was buried or not exposed. In the early Cretaceous, tectonic activity intensified, driven by the movement along normal faults of the rifts, which produced kilometer-scale displacements. During this time, the Llanos Basin experienced notably higher rates of tectonic subsidence, providing evidence for increased tectonism (Sarmiento-Rojas et al., 2006). At the upper end of these large normal faults, kilometer-scale tectonic uplift of rifts shoulders and surface topography may have been

generated (Buiter et al., 2023). Although in another tectonic setting, it can be similar to the case of the East African Rift System, where border faults create a topographic relief up to two kilometers at the edges of the Tanzania Craton (Corti, 2009; Ebinger and Scholz, 2011).

Another recent parallel can be drawn from the Colorado Plateau within the context of the Basin and Range province in the United States. Much like other rifts, the Basin and Range province is bordered by elevated margins, such as the Sierra

Nevada on the western side and the Rocky Mountains along with the Colorado Plateau on the eastern flank (Parsons et al., 2006). The Colorado Plateau stands as a less deformed section of a thick lithosphere, elevated ca. 1.9 km. While there little consensus on the mechanisms behind the plateau uplift (Flowers et al., 2010), thermochronological data suggests that the elevation increase correlated with regional extensional tectonism in adjacent areas (Quigley et al., 2010). We propose that the Rio Negro-Juruena belt likely experienced a similar uplift due to events of tectonic activation of normal faults. This

tectonism created a rift shoulder and an elevated plateau that was rapidly eroded by the increased river erosion power (Fig. 4). The fluvial systems transported the detrital sediments to the Llanos Basin and the Putumayo-Oriente-Maranon sedimentary Province, ultimately resulting in the exhumational cooling of the cratonic area observed in our AFT data (Fig. 4).

The thermal history modeling indicates a gradual decrease in basement cooling rates in the study area from the late

Cretaceous onwards (Fig. 5). Simultaneously, the tectonic subsidence rate of the basins experienced a significant reduction





in the post-Cenomanian (post-95 Ma), suggesting a correlation between the decrease in extensional tectonic activity and the diminishing of exhumation rates and sediment production (Fig. 4; Cooper et al., 1995). The subsidence mechanism changed at this time from fault-induced subsidence to thermal subsidence (Sarmiento-Rojas et al., 2006). Our interpretation suggests that the thermal subsidence process during the post-rift phase resulted in a less elevated Rio Negro-Juruena basement in the late Cretaceous compared to the early Cretaceous. The activity of normal faults, responsible for the uplift, had already ceased. This, in turn, led to a reduction in erosion/denudation and thus exhumation rates.


**Figure 5: Age histograms depicting detrital zircon U-Pb ages for representative sedimentary rocks from the Llanos and Oriente basins. Stratigraphic location of samples shown in Fig. 3. Data from Horton et al., (2010), Guitiérrez et al., (2019) and Vallejo et**





**al., (2021). M-I: Maroni-Itacaiúnas belt; V-T: Ventuari-Tapajós belt; R-J: Rio Negro-Juruena; Rod.: Rondonia belt; GV-S: Greenville-Sunsás; Gond.: Gondwanides orogeny.**

Our data do not reveal evidence of the Cenozoic Andean mountain-building strongly affecting the thermo-tectonic evolution of our study area. The temperature estimates from our samples suggest that they resided in the uppermost crust at in colder conditions, i.e. below the upper APAZ temperature limit (< 60 °C), since the end of the Cretaceous. Thus, there is no

significant thermal overprinting or cooling related to the Cenozoic Andean orogeny in our study area, at least not significant enough to be detectable by the AFT system. This lack of detectable impact might be due to the limitations of the AFT. While our model indicates a mild reheating of about 10 °C around 80 Ma, further investigation is necessary to determine if this event might be real, given the constraints of the AFT method for lower temperatures. In any case, a limited exhumation (ca. 1 km) of the Rio Negro-Juruena belt during the Cenozoic is observed. Detrital sediment provenance data further supports our

findings, showing a substantial reduction in the proportion of basement source rocks older than 450 Ma in Paleogene (ca. 66 Ma) and younger deposits (Fig. 5; Horton, 2018). Instead, there is a distinct presence of ages characteristic of the late Paleozoic-Cenozoic period (post-300 Ma), which are indicative of Andean sediment sources (Fig. 5; Horton et al., 2010; Villagómez et al., 2011; Bustamante et al., 2016). These Andean signatures are closely linked to Cretaceous-Paleogene magmatic arc rocks and Permian-Jurassic intrusive rocks exposed in the retroarc fold-thrust belt. While direct input from the

craton is no longer present, it is worth noting that some recycled cratonic signatures persist in the Paleogene strata. This recycling occurs as old zircon grains are sourced from pre-Andean basin fill within the fold-thrust belt (Martin-Gombojav and Winkler, 2008; Horton et al., 2010; Bande et al., 2012; Horton, 2018). The limited contribution of our study area as a sediment source for the surrounding basins, combined with its current low elevation, suggests that the area has likely experienced Cenozoic subsidence instead of uplift. This subsidence could be attributed to the eastward growth of the

orogenic wedge that induces regional flexure in the foreland domain. As the sedimentary basins migrated eastward over the craton, the study area probably became part of the distal backbulge zone, leading to a gradual sinking or reduction in elevation over the course of the Cenozoic (Horton, 2018; Pachón-Parra et al., 2020). This resulted in lower erosion and exhumation rates of the craton at that time.

## 6.3 Comparative analysis across West Gondwana cratonic segments

Based on the concept of cratonic stability, it would be logical to anticipate an ongoing, gradual process of craton exhumation persisting from the Archean or Proterozoic consolidation until the present time. Such an expectation would predict a shared and comparable exhumation history of cratons, consequently leading to comparable results from apatite fission track (AFT) analyses. However, our study, in conjunction with existing research, reveals that this assumption does not hold true. Here, we present a brief comparative analysis of AFT data from the Ro Negro-Juruena belt with corresponding data from other

cratonic blocks that were once integral parts of the West Gondwana paleocontinent, specifically the São Francisco Craton (SFC), Rio de la Plata Craton (RPC), West African Craton (WAC), Zimbabwe and Kaapvaal Cratons (ZKC), and Tanzania Craton (TC) (Fig. 6).





**Figure 6: Apatite fission track (AFT) age compilation for cratonic segments that once were part of West Gondwana. RTJ:**
**Recôncavo-Tucano-Jatobá aulacogen.**

First we compare our results with those from different regions in the Amazonian Craton. Harman et al. (1998) analysed basement samples along the eastern edge of the Guaporé Shield in Brazil, encompassing the Central Amazonia (> 2.5 Ga) and Maroni-Itacaiúnas (2.2 – 1.95 Ga) provinces (Fig. 1). These samples were categorized into two distinct clusters based on their AFT central ages: the first cluster comprises samples yielding Jurassic to Early Cretaceous (ca. 200 – 100 Ma) central

ages, while the second cluster consists of samples with central ages from the Permian to the Carboniferous (ca. 310 – 250 Ma) (Fig. 6). The samples from the first cluster were collected along the northern border of the shield and exhibit results



similar to ours in terms of AFT central age and mean track length (MTL). In contrast, the samples from the second cluster come from within the interior of the craton. The authors suggest that the differential opening of the equatorial Atlantic Ocean basins was responsible for Meso-Cenozoic exhumation that predominantly impacted the cratonic edge, closer to the rift

zones, signifying that proximity to extensional settings influences cratonic exhumation significantly (Fig. 6). Supporting this assumption, AFT results from Derycke et al. (2021) on four samples from the Guiana Shield align well with the first cluster from Harman et al. (1998) as well as our own findings. These samples yield comparable AFT results, i.e., Jurassic to early Cretaceous central ages and medium to long MTL values (ca. 12 µm), suggesting clear Mesozoic cooling. Notably, these samples come from the passive margin of French Guiana, directly influenced by the rifting of the equatorial Atlantic Ocean.

Initial formation of the Takutu rift in the Amazonian Craton is also linked to the opening of the equatorial Atlantic ocean, rendering this region a comparable geological framework (Fig. 1). However, to date, the basement of this rift system has not been analyzed using low-temperature thermochronometric techniques such as AFT, which prevents direct comparisons with our data.

In the context of the opening of the southern Atlantic Ocean in the South America, both the SFC and the RPC underwent a

process of rifting, resulting in their separation from their corresponding African counterparts of the Congo Craton and the ZKC (Fig. 6). From the SFC, several AFT studies (Harman et al., 1998; Turner et al., 2008; Japsen et al., 2012; Jelinek et al., 2014; Fonseca et al., 2021, 2022) consistently show an intricate pattern of craton exhumation (Fig. 6). Central ages spanning the Mesozoic to Cenozoic are particularly concentrated along the borders of the craton, particularly those regions that experienced the effects of Atlantic rifting, specifically along the northeastern borders (Turner et al., 2008; Japsen et al.,

2012; Jelinek et al., 2014). This age group is also prominent in the vicinity of the Recôncavo-Tucano-Jatobá (RTJ) aulacogen, which evolved during the early Cretaceous (Jelinek et al., 2020). The AFT results suggest that rifting was responsible for km-scale exhumation along the edges of the SFC in the early Cretaceous, potentially through a mechanism akin to the one that triggered the early Cretaceous cooling events in the Amazonian Craton, i.e., a scenario involving the uplift of rift shoulders followed by their subsequent erosion. However, the prevalence of Mesozoic and Cenozoic central

ages diminishes significantly as one moves beyond a distance of 50 km from the border rift faults. These younger AFT ages become notably absent, particularly in the more inland portions of the craton, such as in the state of Minas Gerais in Brazil, where the Archean nucleus of the SFC is situated (Fonseca et al., 2021). In these "protected" areas, Paleozoic ages ranging from the Carboniferous to the Triassic prevail, indicating a reduced level of Mesozoic deformation far from the rift zones (Fig. 6). This pattern is also observed in the Harman et al. (1998) study in the Guaporé Shield, where samples from the

craton interior also yield Paleozoic central ages. The phenomenon likely stems from the structural rigidity inherent in specific segments of some cratons, effectively channeling deformation into narrow, weakened zones, rather than facilitating its widespread regional propagation.

In our study area, along the western border of the Amazonian Craton, Cretaceous deformation appears to have extended over considerable distances in contrast to observations in the SFC. This is evidenced by the fact that all of our samples, even those

from more than 200 km away from the Llanos basin rift show the same thermal history. We hypothesize that the Rio Negro-





Juruena belt underwent a reduction in stiffness over its geodynamic history, facilitating and partitioning more widespread deformation during the early Cretaceous rifting. Our suggestion is rooted also in the observation of the comparatively lower elastic thickness of the Rio Negro-Juruena belt, in contrast to other cratonic segments (Pérez-Gussinyé et al., 2007). Much of the geological history of our study area remains elusive, making it challenging to speculate on events that could have

potentially influenced lithospheric rigidity. The Mesoproterozoic orogenic events that affected the area stand out as plausible candidates. These events were responsible for generating the extensive NW-SE and NE-SW-trending foliation in the granitic rocks, and were associated with elevated temperatures, reaching amphibolite to granulite conditions (e.g., Mendes et al., 2021).

In southeastern South America, basement samples from the RPC are exposed in a small area (< 400 km2) along the

Uruguayan coast. Rocks from the RPC exhibit a wide spectrum of AFT central ages, i.e. Carboniferous to early Cretaceous (ca. 350 – 100 Ma) (Fig. 6; Kollenz et al., 2017; Gomes and Almeida, 2019; Machado et al., 2020). While there are differing perspectives among these authors regarding the extent to which Mesozoic rifting influenced the craton's exhumation, they collectively concur that the exhumation process was not uniform across the entire cratonic region. It is reasoned that faulting during the Mesozoic might have potentially affected certain distinct cratonic sectors, hence controlling their exhumation and

cooling. This observation underscores the ability of the RPC to concentrate deformation in narrow weak zones, similar to the SFC, where a diverse range of AFT ages is localized within relatively small corridors (< 50 km width).

In northwestern Africa, the WAC forms the counterpart of the Amazonian Craton (Fig. 6). The AFT data obtained from both the southern and northern borders of the craton along the current Atlantic coast closely resemble the AFT results published by Derycke et al. (2021) for samples from the Guiana Shield. The AFT data exhibit early Cretaceous to Jurassic central ages

(ca. 200 – 100 Ma) and MTL values of ca. 12 – 13 µm (Oukassou et al., 2012; Fernie et al., 2018; Wildman et al., 2022). The thermal history models unveil episodes of cooling during the early Cretaceous, likely associated with the erosion of rift shoulders resulting from the opening of the Atlantic. Wildman et al. (2022) report a trend of increasing AFT ages with distance from the coast and higher elevation. This phenomenon was attributed by the authors to the specific geomorphology of the passive margin, where the present-day continental drainage divide significantly impacts erosion patterns.

Consequently, the greatest denudation magnitudes are found in the region between the coast and the continental divide, leading to older AFT ages within the continental interior (Gallagher and Brown, 1999). However, in instances where the age difference is more pronounced, such as in Benin (Wildman et al., 2019), the elevation difference does not exceed 400 meters. This suggests that the age difference is not solely due to geomorphology and implies that the magnitude of deformation (uplift) has played a role. The extent of this deformation is likely governed by the pre-existing rheology of the craton.

The ZKC, located in the southern part of Africa, stands out as a significantly elevated craton when compared to most other cratonic regions (ca., 1000 meters; Artemieva and Vinnik, 2016). As a general trend, AFT data from the ZKC follow a pattern similar to the ones previously discussed. Border regions tend to exhibit younger ages, mainly Cretaceous ages (e.g., Wildman et al., 2017; Belton, 2006), while Paleozoic ages are more prevalent toward their centers (e.g., Mackintosh et al., 2017). Stanley et al. (2013, 2015) and Wildman et al. (2017) proposed that the variable denudation patterns observed in





southern Africa, as indicated by AFT data, are likely influenced by a combination of factors. These factors encompass the effects of horizontal plate tectonic stresses at the craton margins, which may have been intensified by changes in plate motions during opening of the South Atlantic Ocean in the Cretaceous, as well as the influence of a lithospheric thermal anomaly. The buoyant upwelling of the mantle could have generated vertical mantle stresses, exerting upward pressure and resulting in the gradual uplift of extensive and robust cratonic interiors. Conversely, areas of weakened lithosphere along the

craton margins might have experienced short-wavelength deformations. The ZKC serves as a pertinent case study illustrating that mantle plumes or variations in the mantle dynamics can induce the surface uplift of cratons. Nonetheless, the extent to which this factor significantly triggers exhumation episodes remains uncertain, especially considering the concurrent occurrence of other geodynamic events such as rifting within the same timeframe. In our study area, there is no evidence suggesting that the Central Atlantic Magmatic Province (CAMP) could have triggered a substantial uplift to form an elevated

plateau during the Jurassic. Contrarily, the provenance analysis of nearby basins does not indicate the Rio Negro-Juruena belt as a significant source area in the Jurassic (Fig. 5), implying that this area was likely at low elevations during that period.

The TC, located in central Africa, constitutes a relatively small and elevated (>1000 m) cratonic fragment  surrounded by high elevated flanks (ca. 1700m) of the East African Rift System (Fig. 6). Tracing its origins back to the Paleogene (35 Ma;

Macgregor et al., 2015), the East African Rift System has undergone continuous evolution, manifesting its structural framework through three primary branches. These branches circumvent the Archaean cratons, as the TC, and predominantly propagate through inherited N-striking orogenic belts (Chorowicz, 2005). This context gives us the opportunity to scrutinize the consequences of active intracontinental rifting adjacent to cratons, offering a modern analogue. A rather limited AFT dataset of the TC exists (Noble et al., 1997; Van Der Beek et al., 1998; Kasanzu, 2017). The central AFT ages in the craton

interior are Carboniferous to Permian (ca. 250 – 350 Ma), indicating a limited degree of potential exhumation linked to Cenozoic rifting processes. Nevertheless, the current substantial elevation of the craton suggests that it has probably experienced uplift, yet significant erosion has not yet taken place. At the boundaries with the East African Rift System, basement samples yield Cretaceous AFT ages. These ages are separated from samples with older ages by major faults (Noble et al., 1997). The authors suggest that exhumation was more intense at the TC cratonic margins, which aligns well with our

prior observations in other cratons.

In summary, low-temperature thermochronology such as AFT analysis reveals that the exhumation of cratons occurred episodically throughout the Phanerozoic. The process of rifting, particularly during the Mesozoic and Cenozoic disassembly of larger continental units in which the cratons are embedded, has exerted a significant influence on craton exhumation. Exhumation patterns are non-uniform across the cratonic regions, showing variability based on proximity to rift zones or

tectonically active boundaries. For instance, in regions such as the SFC and RPC, Mesozoic and Cenozoic ages are clearly concentrated along the craton's borders, especially in areas impacted by Atlantic rifting. In our study area, more widespread Cretaceous exhumation to within the craton's interior is evident. This can most likely be attributed to the weakening of that specific segment of the cratonic lithosphere. In line with the hypothesis proposed by Bedle et al. (2021), our study

underscores the significant role the geological history of the craton and surrounding areas plays in shaping deformation patterns. Events like orogenesis, lithospheric thermal anomalies, and previous tectonic processes can all influence the rigidity and susceptibility of the lithosphere to deformation, ultimately governing the outcomes of the exhumation process.

## 6 Conclusions

This study unveils that the Rio Negro-Juruena belt, situated in the western Guiana Shield (Amazonian Craton) of Colombia, experienced a rapid exhumation event during the early Cretaceous. The AFT central ages range from 79.1 ± 3.2 Ma to 177.1 ± 14.8 Ma with mean track lengths are around 12 µm. The preferred inverse thermal modeling indicates a fast cooling event in the early Cretaceous, result of basement exhumation. This exhumation occurred within a larger context of extensional tectonics, characterized by the presence of Andean subduction related back-arc rifts in the Llanos Basin and the Putumayo-Oriente-Maranon sedimentary Province. Published provenance analyses of sediments from these basins provide compelling evidence supporting the hypothesis that the Rio Negro-Juruena belt served as a significant sediment source during the early Cretaceous. Our interpretation suggests that the tectonic activity associated with the back-arc rifts uplifted the study area, enhancing erosion and ultimately leading to the denudational exhumation of the region.

Moving into the late Cretaceous, the tectonic environment shifted from an extensional to a contractional setting. Basement cooling rates of the Rio Negro-Juruena belt notably decreased, accompanied by diminishing of subsidence patterns in the basins. These changes indicate a decline in tectonic activity in the area, resulting in a reduction in the exhumation of the Rio Negro-Juruena belt. Throughout the Cenozoic, the Rio Negro-Juruena belt likely experienced gradual subsidence and elevation reduction due to the regional flexure induced by the Andean orogeny.

As in our study area, other cratons have been episodically exhumated during the Phanerozoic. Low temperature thermochronology, especially apatite fission track (AFT) analysis, demonstrates that extensional settings associated with rifting of the continental crust can induce events of surface uplift and subsequent erosion. This process results in the exhumation of deeper parts of the cratons. Mesozoic exhumation episodes are preferentially located along the cratonic rifted borders and/or aulacogens (e.g., Recôncavo-Tucano-Jatobá aulacogen, Brazil). The tectonism does not seems to propagate further to the interior of cratons, except in cases where the cratonic lithosphere has been weakened, such as in the Rio Negro-Juruena belt. The capacity of a craton to undergo reworking or deformation seems to hinge on the specific tectonic conditions they have experienced since their Precambrian consolidation.

**Author contribution:**

Ana Fonseca: Conceptualization, Formal analysis, Investigation, Visualization, Writing – original draft preparation

Simon Nachtergaele: Conceptualization, Formal analysis, Validation, Writing – review & editing

Amed Bonilla: Conceptualization, Writing – review & editing





Stijn Dewaele: Conceptualization, Writing – review & editing

Johan De Grave: Conceptualization, Supervision, Resources, Data curation, Writing – review & editing

**Competing interests:**

The authors declare that they have no conflict of interest.

**Acknowledgments**

SN was funded by a PhD scholarship from the Research Foundation Flanders (FWO Vlaanderen) number 1161721N. The
research of ABP was funded by a PhD scholarship from Minciencias (before Colciencias) number 647-2014. Zeze Amaya
and Jose A. Franco are thanked for assistance during sampling.

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
