# Peer review of "Extensional exhumation of cratons: insights from the Early Cretaceous Rio Negro-Juruena belt (Amazonian Craton, Colombia)"

_EGUsphere, 2023_

## Referee Comment (RC1)

**Review of Manuscript egusphere-2023-2113. " Extensional exhumation of cratons: insights from the Early Cretaceous Rio Negro-Juruena belt (Amazonian Craton, Colombia)" by Fonseca et al.**

**Paul Green, Geotrack International; October 2023**

The ms provides new apatite fission track data from a remote area of the South American hinterland, and therefore provides welcome insights into the tectonic evolution of the region which should be of interest to a broad range of readers. I recommend publication but urge the authors to consider carefully the comments provided below and within highlighted passages in the annotated pdf files in revising the ms.

1: A key theme throughout the paper is that whereas cratons are traditionally regarded as characterised by long term stability, the results of this study show that this cratonic region has undergone significant Mesozoic tectonism. While I agree with the authors that the notion of cratonic stability still pervades the earth sciences, a considerable body of evidence has been published over the last 40 years or so to that disproves this notion  The review by Kohn and Gleadow (2019), to which the authors refer, summarises evidence from thermochronology for episodes of exhumation in a number of cratonic regions in which several kilometres of section were eroded. More recently, Green et al. (Earth Science Reviews, 2022, v234) have provided evidence, not only from thermochronology but also from a variety of other approaches (most notably from stratigraphic studies by Sloss and others, from as long ago as 1963!), that cratonic regions have undergone km-scale subsidence and burial prior to exhumation. This implies that much of what was removed during exhumation was a former sedimentary cover, rather than basement. This has clear implications for the results of the study under review, and the authors should consider their conclusions accordingly.

2: Much of the discussion in Section 2 on the geological setting focusses on the region to the west of the study region towards the Pacific margin (Llanos Basin etc), but much of the later discussion is related to Atlantic rifting. The similarity of data from the study region with previous studies along the Atlantic margin suggests a strong link and therefore perhaps some discussion of Atlantic rifting might be useful in Section 2.

3: In Line 287 of the ms the authors claim that it is reasonable to assume that their "MaxLike" solutions "provide a credible approximation to the true cooling history". As noted in the annotated PDF, this places way too much faith in the capabilities of the AFT method. As discussed by Green and Duddy (2020; Earth Science Reviews), the best that can be hoped for from AFT data is to recover the main features of the history (e.g. timing of cooling, rapid or slow cooling), which dominate the measured data. I would suggest the authors write that their solutions "provide a reliable indication of the major aspects of the true history", or similar.

4: The plot of MTL vs FTA in Figure S.41 shows two samples (176 and 180) with distinctly younger ages than the majority, and one sample with an older age (170). The differences in the ages of these samples compared to the majority suggests the possibility of significant differences in thermal history across the region. Looking at the locations of these samples in Figure 3 the two youngest ages fall to the south and the older age to the north, supporting the possibility of different histories. The authors might like to investigate these differences, and perhaps question the validity of modelling the data in all samples combined together.

5: Figure S.42 highlights the presence of a number of samples in which track lengths were not measured. I do not recall seeing any discussion of why this is so, and it should be explained. As noted in the highlighted text in Figure S42, this plot should be presented in terms of AFT age on the y-axis (ordinate) and Dpar on the x-axis (Abscissa), which in the correct way to present effect vs

cause. And such plots are always referred to as y vs x, not the other way round as is the case in this ms.

6: The interpretation of the modelled thermal histories is presented in terms of progressive denudation of the basement region. However, as noted in point 1, above, an increasing body of evidence suggests that real histories may include episodes of burial prior to exhumation, with eroded material being a cover sequence deposited prior to the onset of exhumation. The geological discussion in Section 2 elsewhere highlights former marine incursions in neighbouring regions, which may have also reached the study region and led to deposition. This may or may not be relevant, but the authors should at least consider this in their geological interpretation of their results.

7: Figure 6 potentially provides a useful summary of data from previous studies across the region, but the information therein is quite difficult to discern. I urge the authors to consider improvements to this Figure, perhaps with data shown as coloured datapoints or generalised as bands and also possibly including higher resolution tracing of cratonic outlines.

8: Several of the Figure captions are lacking sufficient information to allow a full understanding of what is presented. Some of these are highlighted in the annotated ms. The most serious case is Figure S1 and similar Figures, where the lower plot is described as "Age vs Age plot of single gran ages". What does this mean? What are the ages plotted on each axis?

9: In each section of the Supplementary document, some text introduction to each section would be welcome, to explain what is presented.

10: Section S.3. Why are no error bars plotted on the MTL values? This should be remedied.

11: Section S4. Again, an introductory text would be welcome here explaining why the selected option were used and how they affect the results.

As mentioned earlier, each of the highlighted sections in the annotated pdf file contains comments requiring attention in revising the ms. If these and the comments provided here can be responded to the revised ms should provide a welcome and useful addition to the literature on the evolution of cratonic regions.

Paul F. Green
Geotrack International
October 2023

Associated documents:
Annotated ms
Annotated Supplementary file

[revised manuscript text omitted]

---

## Referee Comment (RC2)

[referee-annotated manuscript omitted]

---

## Author Comment (AC1)

Reply to the comments on manuscript [Extensional exhumation of cratons: insights from the Early Cretaceous Rio Negro-Juruena belt (Amazonian Craton, Colombia)] by "Fonseca et al."

Dear reviewers,

Thank you for your comments on our manuscript. The comments are encouraging and appear to share our judgement that this study and its results are original and important for the regional geological context with broader implications for similar setting worldwide. All of the comments were very helpful for revising and improving our paper. We have studied these comments carefully and have made corresponding corrections that we hope will meet with your approval. The responses to both reviewers' comments are provided in the attached file. We have been able to incorporate changes to reflect most of the suggestions. If you have any further question, please do not hesitate to contact us. We appreciate the time and effort that both of you, Paul Green and Chiara Amadori, have dedicated to providing your valuable feedback on our manuscript.

Kind regards,

Ana Carolina Liberal Fonseca on behalf of all co-authors

**Comments from Paul Green (Reviewer 1)**

- **General Comment:** The ms provides new apatite fission track data from a remote area of the South American hinterland, and therefore provides welcome insights into the tectonic evolution of the region which should be of interest to a broad range of readers. I recommend publication but urge the authors to consider carefully the comments provided below and within highlighted passages in the annotated pdf files in revising the ms..

  **Response:** We sincerely thank Dr. Paul Green for the constructive criticisms and valuable comments, which were of great help in revising the manuscript. Accordingly, the revised manuscript has been systematically improved. Below, we have replied point-by-point to major and minor comments.

- **Major Comment 1:** A key theme throughout the paper is that whereas cratons are traditionally regarded as characterized by long term stability, the results of this study show that this cratonic region has undergone significant Mesozoic tectonism. While I agree with the authors that the notion of cratonic stability still pervades the earth sciences, a considerable body of evidence has been published over the last 40 years or so to that disproves this notion The review by Kohn and Gleadow (2019), to which the authors refer, summarises evidence from thermochronology for episodes of exhumation in a number of cratonic regions in which several kilometres of section were eroded. More recently, Green et al. (Earth Science Reviews, 2022, v234) have provided evidence, not only from thermochronology but also from a variety of other approaches (most notably from stra☐graphic studies by Sloss and others, from as long ago as 1963!), that cratonic regions have undergone km scale subsidence and burial prior to exhumation. This implies that much of what was removed during exhuma☐on was a former sedimentary cover, rather than basement. This has clear implications for the results of the study under review, and the authors should consider their conclusions accordingly.

**Response:** We appreciate the comment and largely agree with it. The perception of cratonic stability has been gradually shifting over the last 40 year. In our revised manuscript, we've taken your feedback into account, emphasizing these earlier studies that first challenged the traditional notion of cratonic stability (L.34-35). We've incorporated Green et al. (2022) and their findings to the discussions (L.429-432) . Moreover, we have expanded our discussion to encompass potential mechanisms involving burial and erosion, acknowledging that these processes could have implications for understanding the exhumation of cratonic regions (L.367-368). Please, see the response to the Major Comment 6 from the "Comments from Reviewer 1".

- **Major Comment 2:** Much of the discussion in Section 2 on the geological setting focusses on the region to the west of the study region towards the Pacific margin (Llanos Basin etc), but much of the later discussion is related to Atlantic rifting. The similarity of data from the study region with previous studies along the Atlantic margin suggests a strong link and therefore perhaps some discussion of Atlantic rifting might be useful in Section 2.

  **Response:** We thank the reviewer for this suggestion. We value your perspective on the potential triggers for the exhumation observed in our study area. Undoubtedly, during the Early Cretaceous, the activity of surrounding extensional tectonics might have contributed to prompting tectonic activity and subsequent exhumation within the Rio Negro-Juruena belt. However, it is crucial to highlight the closer proximity of our study area to the Llanos and Putumayo-Oriente-Maranon rift system, the parallelism between the directions of extensional strain and the craton structures, and the presence of detrital sediments sourced from our study in the rift basin. These observations strongly indicate that the extension originating from the Andes side was the primary influential factor in driving the exhumation within our study area. At a narrow rifting setting such as the equatorial Atlantic, the significance of proximity to the rift zone is evident, considering that deformation typically remains localized without extending over great distances (e.g., Sapin et al., 2021). While we acknowledge that our interpretation involves a level of speculation, based on available data, we think that attributing the exhumation to the extension from the Andes side remains the most plausible explanation. Perhaps in our original discussion this was not sufficiently clarified by these arguments. To remediate this, we have slightly reformulated section 6.2 to present our arguments for this interpretation more explicitly (L.346-354).

- **Major Comment 3:** In Line 287 of the ms the authors claim that it is reasonable to assume that their "MaxLike" solutions "provide a credible approximaon to the true cooling history". As noted in the annotated PDF, this places way too much faith in the capabilies of the AFT method. As discussed by Green and Duddy (2020; Earth Science Reviews), the best that can be hoped for from AFT data is to recover the main features of the history (e.g. timing of cooling, rapid or slow cooling), which dominate the measured data. I would suggest the authors write that their solutions "provide a reliable indication of the major aspects of the true history", or similar.

**Response:** Agree. As suggested by the reviewer, we have amended our manuscript, rewriting the sentence as suggested (L.313).

- **Major Comment 4:** The plot of MTL vs FTA in Figure S.41 shows two samples (176 and 180) with distinctly younger ages than the majority, and one sample with an older age (170). The differences in the ages of these samples compared to the majority suggests the possibility of significant differences in thermal history across the region. Looking at the locaons of these samples in Figure 3 the two youngest ages fall to the south and the older age to the north, supporting the possibility of different histories. The authors might like to investigate these differences, and perhaps question the validity of modelling the data in all samples combined together.

  **Response:** We appreciate your observations regarding the variations in the AFT age evident in Figure S.41 for samples 176, 180, and 170. You are correct in noting the divergence of these samples as compared to the majority of the samples. This is indeed also depicted in the Observed vs. Predicted graphs in Supplement 4, figures S.43 and S.44. Our intention in modeling these samples jointly was to not only capture the general trend but also to underscore and identify such outliers within the dataset. We regret this was not adequately addressed in the text. Your feedback has prompted us to incorporate this aspect into the revised manuscript (L.288-293). However, upon further consideration the differential thermal histories of these samples presents challenges. For instance, surrounding samples such as 175, 182, and 171 do not substantiate differential reactivations. As such, we will mention the age variation in the revised manuscript but we will not extensively discuss this aspect at this stage.

- **Major Comment 5:** Figure S.42 highlights the presence of a number of samples in which track lengths were not measured. I do not recall seeing any discussion of why this is so, and it should be explained. As noted in the highlighted text in Figure S42, this plot should be presented in terms of AFT age on the y-axis (ordinate) and Dpar on the x-axis (Abscissa), which in the correct way to present effect vs cause. And such plots are always referred to as y vs x, not the other way round as is the case in this ms.

  **Response:** We apologize for any confusion, figure S.42 illustrates the relationship between $D_{par}$ and AFT central ages, encompassing samples where an (in)sufficient number of confined tracks were measured. Indeed, the parameters displayed in the x and y axis should be interchanged. We have, accordingly, adequate the graph.

- **Major Comment 6:** The interpretation of the modelled thermal histories is presented in terms of progressive denudation of the basement region. However, as noted in point 1, above, an increasing body of evidence suggests that real histories may include episodes of burial prior to exhumation, with eroded material being a cover sequence deposited prior to the onset of exhumation. The geological discussion in Section 2 elsewhere highlights former marine incursions in neighbouring regions, which may have also reached the study region and led to deposition. This may or may not be relevant, but the authors should at least consider this in their geological interpretation of their results.

**Response:** We agree that that part of the material that was removed during the early Cretaceous might represent Paleozoic sediments and/or sedimentary rocks. We have added that possibility to the text (L.367-368).

- **Major Comment 7:** Figure 6 potentially provides a useful summary of data from previous studies across the region, but the information therein is quite difficult to discern. I urge the authors to consider improvements to this Figure, perhaps with data shown as coloured datapoints or generalised as bands and also possibly including higher resolution tracing of cratonic outlines.

  **Response:** Thank you for your feedback regarding Figure 6. We appreciate your suggestion to use colored dots for discrete sample points to illustrate the age variation across the region. However, we wanted to shed light on the practical challenges we encountered in implementing this suggestion. Georeferencing a dataset as extensive as that one, with over a thousand samples, is undoubtedly an ideal approach. Unfortunately, this make the figure too crowded and confusing because a lot of points overlap in the scale of the figure. Thus, we had to prioritize demonstrating the primary trends within the data rather than achieving detailed georeferencing. Our intention was to offer a broad overview of the main data trends, which we think the current figure adequately portrays. Additionally, for readers interested in accessing the dataset's specifics, the figure includes guidance for acquiring the necessary data from the original paper. While we agree that a more detailed georeferencing would be beneficial, we think the current figure effectively serves the purpose of this paper.

- **Major Comment 8:** Several of the Figure captions are lacking sufficient information to allow a full understanding of what is presented. Some of these are highlighted in the annotated ms. The most serious case is Figure S1 and similar Figures, where the lower plot is described as "Age vs Age plot of single gran ages". What does this mean? What are the ages plotted on each axis?

  **Response:** The Age vs. Age plot is a graphical representation of single grain age dispersion, where the central age of each grain is plotted against the central age with the uncertainty displayed on the y-axis. After reevaluating the relevance of this graph, we have decided to remove them, as it is not deemed entirely necessary within this context. We consider that the radial plots do better in demonstrating the dispersion and uncertainty.

- **Major Comment 9:** In each section of the Supplementary document, some text introduction to each section would be welcome, to explain what is presented.

  **Response:** We thank the reviewer for this feedback. We have, accordingly, added a brief introduction to each section of the Supplement.

- **Major Comment 10:** Section S.3. Why are no error bars plotted on the MTL values? This should be remedied.

**Response:** We thank the reviewer for highlighting this point. We have, accordingly, added the error bars plotted on the MTL values.

- **Major Comment 11:** Section S4. Again, an introductory text would be welcome here explaining why the selected option were used and how they affect the results.

  **Response:** We thank the reviewer for this feedback. We have, accordingly, added a brief introduction to each section of the Supplement.

- **Minor Comment 1:** Why is this likely? We are in the centre of a cratonic region so the impetus for exhumation could just as likely come from the south or east, particularly as similar age exhumation has been reported in those areas. **(Abstract)**

  **Response:** Please, see the response the Major Comment 2 from the "Comments from Reviewer 1".

- **Minor Comment 2:** Why controversial? **(Introduction)**

  **Response:** Thank you for pointing this out. We have removed this adjective from the text because it is unnecessary and might cause misunderstandings.

- **Minor Comment 3:** SO what did they find? **(Introduction)**

  **Response:** We regret that our original statement did not make clear the finds of Bonilla et al., (2020). We have, accordingly, added this information in the text (L.76-79).

- **Minor Comment 4:** define precisely and refer to Fig 3. **(Geological Setting)**

  **Response:** Agree. We have modified the sentence and referred to the figure (L.111).

- **Minor Comment 5:** incorrect useage. Simply Ga, which refers to a geological time. **(Geological Setting)**

  **Response:** Thank you for pointing this out. We removed the "ago" which was incorrect.

- **Minor Comment 6:** Given the evidence for marine transgressions and continental deposition, isn't it likely that your basement samples were covered by sediment at some time during the {Phanerozoic? **(Geological Setting)**

  **Response:** While we understand the reviewer's perspective, and we acknowledge the possibility that basement samples might have been covered by sediment during the Phanerozoic. However, given the absence of geological evidence, such as remnants of these proposed sediments, we think that addressing this discussion in the subsequent section (6.2) is more appropriate. This section provides ample space for a thorough

analysis of hypotheses. We have included a brief section regarding this matter (L.367-368). Please, see also the response to the Major Comment 6 from the "Comments from Reviewer 1".

- **Minor Comment 7:** This isn't really accurate. The mesning of an apatite fission track age is much more complex. **(Samples and methods)**

  **Response:** We appreciate the comment and largely agree with it. We have modified the sentence to include some of the complexities regarding the meaning of AFT age (L.175-177).

- **Minor Comment 8:** Again not really this simple. **(Samples and methods)**

  **Response:** While we understand the reviewer's point of view and acknowledge that the apatite partial annealing zone (APAZ) is not fully constrained or equal for all apatites, we consider the APAZ concept important in evaluating and interpreting AFT data in a first order. Despite its complexities, this concept remains widely used and advocated (see, for instance, the review work by Malusà et al., 2019). Hence, we have chosen to retain it within the text. To avoid oversimplification, we have added a sentence regarding the influence of apatite composition and cooling rate on annealing rates and consequently AFT age (L.182-184).

- **Minor Comment 9:** an outmoded concept. **(Samples and methods)**

  **Response:** See the response to the Minor Comment 8 from the "Comments from Reviewer 1".

- **Minor Comment 10:** This is a mistake becaiuse AFT modelling software is based on length bias calculations for horizontal tracks only. **(Samples and methods)**

  **Response:** While we appreciate the reviewer's feedback, we disagree to some extent with his comment. Indeed, the modeling software relies on length measurements primarily measured for horizontal tracks due to limitations in some equipment's precision for determining track inclinations accurately. Consequently, inclined tracks are often disregarded because their actual lengths cannot be precisely determined. Nonetheless, our microscope and software set-up capability to transform projected lengths of sub-horizontal tracks to their true lengths allows us to measure more confined tracks effectively. Our observations reveal that despite this correction, there is only a small increase of 0.4 μm (3%) at most, owing to the slight inclinations of the tracks, as evidenced in the table below. Based on these findings, we are convinced that correcting for inclination does not significantly impact the modeling or its interpretations.

Reply to the comments on manuscript [Extensional exhumation of cratons: insights from the Early Cretaceous Rio Negro-Juruena belt (Amazonian Craton, Colombia)] by "Fonseca et al."

| 506 | length | la | angle c-axis | corrected angle (0-90) | z1 | z2 | dz | corrected inclination | corrected (-) uncorrected | type | track | length | la | angle c-axis | corrected angle (0-90) | z1 | z2 | dz | corrected inclination | corrected (-) uncorrected | type |
|---|---|---|---|---|---|---|---|---|---|---|---|---|---|---|---|---|---|---|---|---|---|
| 1 | 13,7 | 14,0 | 163,0 | 17,0 | 37,3 | 36,5 | 0,8 | 14,1 | 0,1 | TINT | 51 | 13,8 | 14,2 | 326,0 | 34,0 | 74,9 | 74,3 | 0,6 | 14,2 | 0,0 | TINT |
| 2 | 9,8 | 10,0 | 242,0 | 62,0 | 36,2 | 36,2 | 0,0 | 10,0 | 0,0 | TINT | 52 | 14,0 | 14,4 | 311,0 | 49,0 | 75,2 | 74,7 | 0,5 | 14,4 | 0,0 | TINT |
| 3 | 9,1 | 9,3 | 120,0 | 60,0 | 36,1 | 36,1 | 0,0 | 9,3 | 0,0 | TINT | 53 | 11,6 | 11,8 | 35,0 | 35,0 | 70,1 | 69,7 | 0,4 | 11,9 | 0,0 | TINT |
| 4 | 7,3 | 7,5 | 142,0 | 38,0 | 37,1 | 36,2 | 0,9 | 7,6 | 0,2 | TINT | 54 | 13,3 | 13,6 | 340,0 | 20,0 | 70,5 | 69,2 | 1,3 | 13,8 | 0,2 | TINT |
| 5 | 11,6 | 11,9 | 228,0 | 48,0 | 60,2 | 59,3 | 0,9 | 12,0 | 0,1 | TINT | 55 | 14,4 | 14,8 | 310,0 | 50,0 | 69,1 | 69,6 | -0,5 | 14,8 | 0,0 | TINT |
| 6 | 8,4 | 8,6 | 301,0 | 59,0 | 59,4 | 59,4 | 0,0 | 8,6 | 0,0 | TINT | 56 | 11,5 | 11,7 | 277,0 | 83,0 | 69,3 | 69,9 | -0,6 | 11,8 | 0,0 | TINT |
| 7 | 11,5 | 11,7 | 262,0 | 82,0 | 60,6 | 60,1 | 0,5 | 11,8 | 0,0 | TINT | 57 | 13,0 | 13,3 | 240,0 | 60,0 | 70,5 | 69,7 | 0,8 | 13,3 | 0,1 | TINT |
| 8 | 9,7 | 10,0 | 131,0 | 49,0 | 69,2 | 69,7 | -0,4 | 10,0 | 0,0 | TINT | 58 | 12,3 | 12,6 | 306,0 | 54,0 | 68,8 | 68,8 | 0,0 | 12,6 | 0,0 | TINT |
| 9 | 13,9 | 14,2 | 208,0 | 28,0 | 69,2 | 68,3 | 1,0 | 14,3 | 0,1 | TINT | 59 | 12,5 | 12,8 | 233,0 | 53,0 | 65,8 | 64,3 | 1,5 | 13,0 | 0,2 | TINT |
| 10 | 11,9 | 12,2 | 141,0 | 39,0 | 69,1 | 68,3 | 0,8 | 12,2 | 0,1 | TINT | 60 | 11,6 | 11,9 | 220,0 | 40,0 | 65,9 | 65,9 | 0,0 | 11,9 | 0,0 | TINT |
| 11 | 10,9 | 11,2 | 131,0 | 49,0 | 69,5 | 67,6 | 1,9 | 11,6 | 0,4 | TINT | 61 | 14,2 | 14,6 | 286,0 | 74,0 | 65,4 | 65,8 | -0,4 | 14,6 | 0,0 | TINT |
| 12 | 13,4 | 13,7 | 321,0 | 39,0 | 67,3 | 67,3 | 0,0 | 13,7 | 0,0 | TINT | 62 | 12,1 | 12,4 | 303,0 | 57,0 | 66,1 | 65,2 | 0,9 | 12,5 | 0,1 | TINT |
| 13 | 10,7 | 11,0 | 26,0 | 26,0 | 68,0 | 68,0 | 0,0 | 11,0 | 0,0 | TINT | 63 | 12,5 | 12,8 | 291,0 | 69,0 | 66,0 | 66,4 | -0,3 | 12,8 | 0,0 | TINT |
| 14 | 14,8 | 15,2 | 216,0 | 36,0 | 68,0 | 67,9 | 0,1 | 15,2 | 0,0 | TINT | 64 | 11,2 | 11,5 | 303,0 | 57,0 | 65,4 | 65,4 | 0,0 | 11,5 | 0,0 | TINT |
| 15 | 12,0 | 12,3 | 161,0 | 19,0 | 68,8 | 69,0 | -0,2 | 12,3 | 0,0 | TINT | 65 | 14,0 | 14,3 | 334,0 | 26,0 | 64,9 | 64,9 | 0,0 | 14,3 | 0,0 | TINT |
| 16 | 13,8 | 14,1 | 291,0 | 69,0 | 69,1 | 68,6 | 0,5 | 14,1 | 0,0 | TINT | 66 | 12,6 | 12,9 | 291,0 | 69,0 | 59,7 | 59,2 | 0,5 | 12,9 | 0,0 | TINT |
| 17 | 11,8 | 12,1 | 301,0 | 59,0 | 72,0 | 71,0 | 1,0 | 12,2 | 0,1 | TINT | 67 | 13,1 | 13,4 | 244,0 | 64,0 | 62,0 | 61,1 | 0,8 | 13,4 | 0,1 | TINT |
| 18 | 11,0 | 11,3 | 83,0 | 83,0 | 71,6 | 72,1 | -0,5 | 11,3 | 0,0 | TINT | 68 | 11,5 | 11,8 | 264,0 | 84,0 | 60,9 | 60,8 | 0,1 | 11,8 | 0,0 | TINT |
| 19 | 13,7 | 14,1 | 45,0 | 45,0 | 72,0 | 71,6 | 0,5 | 14,1 | 0,0 | TINT | 69 | 13,0 | 13,3 | 286,0 | 74,0 | 61,5 | 61,7 | -0,2 | 13,3 | 0,0 | TINT |
| 20 | 14,3 | 14,6 | 319,0 | 41,0 | 72,0 | 71,6 | 0,4 | 14,7 | 0,0 | TINT | 70 | 10,4 | 10,7 | 262,0 | 82,0 | 72,0 | 70,9 | 1,1 | 10,8 | 0,1 | TINT |
| 21 | 12,1 | 12,4 | 294,0 | 66,0 | 76,5 | 76,0 | 0,5 | 12,4 | 0,0 | TINT | 71 | 10,3 | 10,5 | 327,0 | 33,0 | 70,8 | 70,5 | 0,3 | 10,5 | 0,0 | TINT |
| 22 | 12,3 | 12,6 | 302,0 | 58,0 | 75,9 | 76,3 | -0,4 | 12,6 | 0,0 | TINT | 72 | 14,0 | 14,3 | 331,0 | 29,0 | 72,5 | 72,7 | -0,1 | 14,3 | 0,0 | TINT |
| 23 | 10,8 | 11,1 | 182,0 | 2,0 | 76,1 | 75,2 | 0,8 | 11,2 | 0,1 | TINT | 73 | 11,2 | 11,5 | 305,0 | 55,0 | 72,0 | 71,1 | 0,9 | 11,6 | 0,1 | TINT |
| 24 | 13,7 | 14,0 | 285,0 | 75,0 | 76,7 | 76,8 | -0,1 | 14,0 | 0,0 | TINT | 74 | 10,6 | 10,9 | 314,0 | 46,0 | 71,5 | 70,9 | 0,6 | 10,9 | 0,0 | TINT |
| 25 | 10,8 | 11,0 | 253,0 | 73,0 | 75,3 | 74,8 | 0,5 | 11,1 | 0,0 | TINT | 75 | 13,4 | 13,7 | 46,0 | 46,0 | 74,0 | 73,5 | 0,5 | 13,8 | 0,0 | TINT |
| 26 | 12,1 | 12,4 | 250,0 | 70,0 | 45,0 | 44,6 | 0,4 | 12,4 | 0,0 | TINT | 76 | 13,3 | 13,6 | 40,0 | 40,0 | 74,7 | 73,4 | 1,3 | 13,8 | 0,2 | TINT |
| 27 | 14,8 | 15,2 | 323,0 | 37,0 | 45,6 | 44,9 | 0,8 | 15,3 | 0,1 | TINT | 77 | 11,1 | 11,4 | 21,0 | 21,0 | 75,2 | 74,7 | 0,5 | 11,4 | 0,0 | TINT |
| 28 | 12,4 | 12,7 | 246,0 | 66,0 | 70,5 | 70,7 | -0,2 | 12,7 | 0,0 | TINT | 78 | 12,7 | 13,0 | 249,0 | 69,0 | 74,1 | 73,1 | 1,0 | 13,1 | 0,1 | TINT |
| 29 | 14,4 | 14,7 | 215,0 | 35,0 | 70,0 | 69,1 | 1,0 | 14,8 | 0,1 | TINT | 79 | 13,1 | 13,4 | 152,0 | 28,0 | 74,9 | 74,3 | 0,6 | 13,4 | 0,0 | TINT |
| 30 | 14,6 | 15,0 | 254,0 | 74,0 | 70,0 | 69,3 | 0,7 | 15,0 | 0,0 | TINT | 80 | 14,0 | 14,3 | 242,0 | 62,0 | 75,2 | 74,7 | 0,5 | 14,3 | 0,0 | TINT |
| 31 | 12,4 | 12,7 | 257,0 | 77,0 | 70,4 | 69,9 | 0,6 | 12,8 | 0,0 | TINT | 81 | 11,7 | 12,0 | 282,0 | 78,0 | 70,1 | 69,7 | 0,4 | 12,0 | 0,0 | TINT |
| 32 | 14,0 | 14,3 | 337,0 | 23,0 | 69,9 | 70,1 | -0,2 | 14,3 | 0,0 | TINT | 82 | 12,2 | 12,5 | 290,0 | 70,0 | 70,5 | 69,2 | 0,5 | 12,5 | 0,0 | TINT |
| 33 | 8,2 | 8,4 | 230,0 | 50,0 | 70,5 | 70,1 | 0,5 | 8,4 | 0,0 | TINT | 83 | 11,7 | 12,0 | 285,0 | 75,0 | 69,1 | 69,6 | -0,5 | 12,0 | 0,0 | TINT |
| 34 | 11,6 | 11,9 | 204,0 | 24,0 | 70,0 | 69,5 | 0,5 | 11,9 | 0,0 | TINT | 84 | 12,9 | 13,2 | 238,0 | 58,0 | 69,3 | 69,9 | -0,6 | 13,2 | 0,0 | TINT |
| 35 | 12,2 | 12,5 | 292,0 | 68,0 | 71,6 | 72,6 | -1,0 | 12,6 | 0,1 | TINCLE | 85 | 9,0 | 9,2 | 238,0 | 58,0 | 62,0 | 61,1 | 0,8 | 9,3 | 0,1 | TINT |
| 36 | 12,8 | 13,2 | 151,0 | 29,0 | 72,7 | 72,0 | 0,7 | 13,2 | 0,0 | TINT | 86 | 14,3 | 14,7 | 161,0 | 19,0 | 60,9 | 60,8 | 0,1 | 14,7 | 0,0 | TINT |
| 37 | 11,6 | 11,9 | 251,0 | 71,0 | 72,0 | 72,0 | 0,0 | 11,9 | 0,0 | TINT | 87 | 12,4 | 12,7 | 286,0 | 74,0 | 61,5 | 61,7 | -0,2 | 12,7 | 0,0 | TINT |
| 38 | 14,1 | 14,4 | 280,0 | 80,0 | 70,9 | 71,3 | -0,5 | 14,4 | 0,0 | TINT | 88 | 12,1 | 12,4 | 74,0 | 74,0 | 72,0 | 70,9 | 1,1 | 12,5 | 0,1 | TINT |
| 39 | 11,6 | 11,8 | 240,0 | 60,0 | 72,0 | 71,3 | 0,7 | 11,9 | 0,1 | TINT | 89 | 12,7 | 13,0 | 76,0 | 76,0 | 70,8 | 70,5 | 0,3 | 13,0 | 0,0 | TINT |
| 40 | 9,6 | 9,9 | 297,0 | 63,0 | 72,0 | 72,0 | 0,0 | 9,9 | 0,0 | TINT | 90 | 9,7 | 9,9 | 38,0 | 38,0 | 72,5 | 72,7 | -0,1 | 9,9 | 0,0 | TINT |
| 41 | 11,4 | 11,7 | 311,0 | 49,0 | 72,0 | 72,0 | 0,0 | 11,7 | 0,0 | TINT | 91 | 12,6 | 12,9 | 303,0 | 57,0 | 69,2 | 69,7 | -0,4 | 12,9 | 0,0 | TINT |
| 42 | 10,4 | 10,7 | 219,0 | 39,0 | 72,0 | 70,9 | 1,1 | 10,8 | 0,1 | TINT | 92 | 11,4 | 11,6 | 60,0 | 60,0 | 69,2 | 68,3 | 1,0 | 11,8 | 0,1 | TINT |
| 43 | 15,8 | 16,2 | 253,0 | 73,0 | 70,8 | 70,5 | 0,3 | 16,2 | 0,0 | TINT | 93 | 13,2 | 13,6 | 51,0 | 51,0 | 69,1 | 68,3 | 0,8 | 13,6 | 0,1 | TINT |
| 44 | 10,8 | 11,1 | 250,0 | 70,0 | 72,5 | 72,7 | -0,1 | 11,1 | 0,0 | TINT | 94 | 10,1 | 10,4 | 293,0 | 67,0 | 69,5 | 67,6 | 1,9 | 10,8 | 0,4 | TINT |
| 45 | 12,3 | 12,6 | 243,0 | 63,0 | 72,0 | 71,1 | 0,9 | 12,7 | 0,1 | TINT | 95 | 13,5 | 13,8 | 252,0 | 72,0 | 67,3 | 67,3 | 0,0 | 13,8 | 0,0 | TINT |
| 46 | 12,7 | 13,0 | 211,0 | 31,0 | 71,5 | 70,9 | 0,6 | 13,0 | 0,0 | TINT | 96 | 13,4 | 13,8 | 317,0 | 43,0 | 68,0 | 68,0 | 0,0 | 13,8 | 0,0 | TINT |
| 47 | 10,1 | 10,3 | 300,0 | 60,0 | 74,0 | 73,5 | 0,5 | 10,3 | 0,0 | TINT | 97 | 13,9 | 14,2 | 277,0 | 83,0 | 68,0 | 67,9 | 0,1 | 14,2 | 0,0 | TINT |
| 48 | 9,4 | 9,7 | 280,0 | 80,0 | 74,7 | 73,4 | 1,3 | 9,9 | 0,2 | TINT | 98 | 13,3 | 13,6 | 337,0 | 23,0 | 68,8 | 69,0 | -0,2 | 13,6 | 0,0 | TINT |
| 49 | 11,5 | 11,8 | 61,0 | 61,0 | 75,2 | 74,7 | 0,5 | 11,8 | 0,0 | TINT | 99 | 14,4 | 14,8 | 345,0 | 15,0 | 72,0 | 70,9 | 1,1 | 14,9 | 0,1 | TINT |
| 50 | 11,5 | 11,8 | 284,0 | 76,0 | 74,1 | 73,1 | 1,0 | 11,9 | 0,1 | TINT | 100 | 12,7 | 13,0 | 65,0 | 65,0 | 70,8 | 70,5 | 0,3 | 13,0 | 0,0 | TINT |

Table for length measurements, sample 506. "Length" represent the raw measurement. "la" is the length corrected by the analyst factor, calculated after measuring a high precision scale for 50 times, in this case the factor is 1.02 (2% correction). "angle with c-axis" includes values from 0 to 360°. "Correct angle" converts the angle with c-axis to values between 0 and 90°. "Z1" and "z2" are the reference elevation value for the microscope when it focus on both ends of the confined track. "Dz" is the difference between z1 and z2. "Corrected inclination" is length value after correcting for inclination using Pythagoras theorem. "Corrected (-) uncorrected" is the difference between the "la" length values and the "corrected inclination" values.

- **Minor Comment 11:** Refer to Figure S.43 **(Thermal History Modelling)**

  **Response:** Agree. We have referred to the figure.

- **Minor Comment 12:** Figure S.44 **(Thermal History Modelling)**

Reply to the comments on manuscript [Extensional exhumation of cratons: insights from the Early Cretaceous Rio Negro-Juruena belt (Amazonian Craton, Colombia)] by "Fonseca et al."

**Response:** Agree. We have, accordingly, referred to the figure which is now part of the manuscript Fig.4. Please, see also the response the Minor Comment 31 from the "Comments from Reviewer 2".

- **Minor Comment 13:** The constraint shown in the Figure appears to be at a temperature of around 60C. Why was this value chsen, when a lower value would be more appropriate for surface exposure? **(Thermal History Modelling)**

  **Response:** Thank you for your suggestion. We opted for the constraint (200±20 Ma, 50±5°C) as a hypothetical low temperature scenario that the basement might have experienced. Acknowledging your point regarding lower temperatures being more suitable for surface exposure, we generated an additional model using the constraint (200±20 Ma, 30±5°C). The results shown below indicate a close alignment, and as we think the primary discussion remains unchanged, we opted not to make alterations to the text.

[Figure]

- **Minor Comment 14:** A more important factor is the degee to which the predictions of these models match the measured data. Perhaps this comes later, but if so it should certainly be mentioned here. If not, then ot should be discussed in detail. **(Thermal History Modelling)**

  **Response:** Agree. We have added the that information to section 5 (L. 244-245).

- **Minor Comment 15:** The manipulation to c-axis projected lengths is not a "correction". It is supposed to render te data more reproducible (although this is debatable). **(Thermal History Modelling)**

  **Response:** Agree. We have replaced the word for "projection" (L.263).

- **Minor Comment 16:** The plots in S3 all show exact;y te form of anisotropy typical of the track lengths in these samples. The pattern only becomre really obvious at high degrees of shortening. **(Thermal History Modelling)**

  **Response:** We regret that we referred to the wrong section of the Supplement. In fact, we meant section S2, where the raw length data for each individual sample does not exhibit

the predicted anisotropy for orientation. We have, accordingly, modified the reference to the correct section (L.268).

- **Minor Comment 17:** This should have been discusssed in Section 5. **(Discussions)**

  **Response:** Agree. Please, see the response to the Minor Comment 14 from the "Comments from Reviewer 1".

- **Minor Comment 18:** More specific details required. **(Discussions)**

  **Response:** Thank you for pointing this out. We have now referred to the specific figure (L.311).

- **Minor Comment 19:** I think this is too optimistic. At best the thermal history solutions can only provide a broad representation of the true hisyory, which is likely to be much more complex. **(Discussions)**

  **Response:** Please, see the response to the Major Comment 3 from the "Comments from Reviewer 1".

- **Minor Comment 20:** see earlier comments. **(Discussions)**

  **Response:** Please, see the response to the Minor Comments 15 and 16 from the "Comments from Reviewer 1".

- **Minor Comment 21:** This is not true. Various authors have attributed Late Cretaceous-Early Cenozoic cooling in the UK to the effects of the Iceland plume. And Japsen et al (2023 Gondwana Research) ) reported exhumation of similar age in Northern Greenland and adjacent regions to a similar process. **(Discussions)**

  **Response:** Thank you for pointing this out. We have, accordingly, deleted these sentences.

- **Minor Comment 22:** Figure 5 shows that before the Early Cretaceous, Jurassic rocks are dominated by ~200 Ma zircons, resumably related to CAMP activity. So the highlighted statement is misleading. The only available comparison with pre-Jurassic rocks is in the Llanos Basin, where the older rocks show a similar pattern to the Early Cretaceous samples. So I do not find the argument presented in the text to be convincing. **(Discussions)**

  **Response:** While we appreciate the reviewer's feedback, we hold a differing view on some aspects. Firstly, it is imperative to clarify that the interpretation of the provenance data outlined in this paragraph is drawn directly from the original papers (see references). Secondly, the reviewer's suggestion about ~200 Ma zircons originating from CAMP activity appears less probable, although it cannot be ruled out completely. CAMP rocks typically

comprise basaltic lava flows and basic intrusions (Marzoli et al., 2018, Davies et al., 2017), which usually yield minimal to no zircons. Moreover, these rocks are situated at a considerable distance from the basins of up to more than 500 km. A more plausible source for these zircons is the adjacent Permian-Jurassic intrusive rocks in the retroarc fold-thrust belt of the Andean orogen, as previously outlined by several studies (e.g., Bustamante et al., 2016). The reviewer rightly highlights that the Paleozoic rocks in the Llanos basin received zircons from the cratonic area. However, it is important to note that most of these ages do not align with those of the Rio Negro-Juruena belt (Fig. 6). To prevent any misunderstandings, we have clarified the sentence to specifically refer to the Jurassic deposits, as these exhibit a minimal contribution from the craton (L.356). These Jurassic rocks primarily yield Phanerozoic zircons.

We reiterate the findings from several studies (Guitiérrez et al., 2019; Vallejo et al., 2019; Guerrero et al., 2020; Horton et al., 2010; Hurtado et al., 2018; Cardona et al., 2010) demonstrating that Jurassic provenance histories were primarily influenced by local Andean sources, displaying significant spatial variability. Conversely, Cretaceous provenance indicates an increased incorporation of regional cratonic sources, with a progressively greater contribution of basement signatures from the Guyana shield, notably including the Rio Negro-Juruena belt.

- **Minor Comment 23:** What evidence do you have for this interpretation? You might also like to consider the nature of the rock that was removed during early Cretaceous exhumation. Was this more basement similar to what is preserved, or was it part of a sedimentary cover, possibly Paleozoic? **(Discussions)**

  **Response:** Please, see the response to the Major Comments 6 from the "Comments from Reviewer 1".

- **Minor Comment 24:** As above, was this because of a sedimentary cover? Your results suggest that a significant thickness of rock was removed during the Early Cretaceous, probably 2 to 3 km. How likely is it that this was simply more of the same rock that is now preserved? **(Discussions)**

  **Response:** Please, see the response to the Major Comments 6 from the "Comments from Reviewer 1".

- **Minor Comment 25:** Exactly. So any changes in provenance do not necessarily reflect the early Cretaceous exhumation. And if the region was covered, then prior to the cover being deposited the underlying basement must have been close to the surface. This changes the nature of the history to one involving burial followed by exhumation. This shoud be considered further. **(Discussions)**

  **Response:** We regret that we did not consider this possibility before. We have added this hypothesis in the discussion. Please, see also the response to the Major Comments 6 from the "Comments from Reviewer 1".

Reply to the comments on manuscript [Extensional exhumation of cratons: insights from the Early Cretaceous Rio Negro-Juruena belt (Amazonian Craton, Colombia)] by "Fonseca et al."

- **Minor Comment 26:** I assume that one column represents the Llanos basin and the other represents the Oriente basin. But which is which? Please label them accordingly. **(Fig. 5)**

  **Response:** Thank you for your comment. The labeling for the Llanos basin and the Oriente basin can now be found at the top of each respective graph in Figure 6.

- **Minor Comment 27:** Although this concept does hold true in many minds, there is now abundant evidence that as a generalisation it is far from true. The paper by Kohn And Gleadow referred to earlier sets out plenty of evidence for episodes of cratonic exhumation throughout the Phanerozoic. More recently, Green et al (2023, Earth Science reviews) have shown that many cratonic regions (as well as other supposed "non-tectonic" settings) ave undergone repeated episodes of burial and exhumation. The authors should take note of this evidence in forming the discussion here. **(Discussions)**

  **Response:** Please, see the response to the Major Comment 1 from the "Comments from Reviewer 1".

- **Minor Comment 28:** This needs larger text in the Figure. **(Fig. 6)**

  **Response:** Agree. We have, accordingly, enlarged the text in the figure.

- **Minor Comment 29:** I think tis Figure needs improvement. It could potentially be a valuable source for future reference but the age variation across the region needs to be dispalyed more clearly, possibly in the form of coloured dots for discrete sample points. As it is, the Figure disguises the invormation rather than illustrating it. **(Fig. 6)**

  **Response:** Please, see the response to the Major Comment 7 from the "Comments from Reviewer 1".

- **Minor Comment 30:** Explain the colour coding of the symbols. **(Fig. 6)**

  **Response:** Agree. We have explained the colour coding of the symbols.

- **Minor Comment 31:** results from Daly et al 2020 (https://doi.org/10.1029/2019GC008746) would also be relevant here. **(Fig. 6)**

  **Response:** We appreciate the suggestion and largely agree with it. We regret that we did not cite this work appropriately. We have added this reference and its discussion to the text (L.511) and to the Figure 7.

- **Minor Comment 32:** exhumed. **(Conclusions)**

  **Response:** Thank you for pointing this out. We have made the correction.

Reply to the comments on manuscript [Extensional exhumation of cratons: insights from the Early Cretaceous Rio Negro-Juruena belt (Amazonian Craton, Colombia)] by "Fonseca et al."

- **Minor Comment 33:** what about removal of former cover? **(Conclusions)**

  **Response:** While we have now included this hypothesis in the Discussion section (Please, see the response to the Major Comment 6 from the "Comments from Reviewer 1") , we remain cautious about integrating it into the paper's conclusion. This caution arises from the lack of additional independent evidence supporting the existence of substantial sedimentary rocks covering the craton basement during the Phanerozoic.

- **Minor Comment 34:** might this to some extent reflect the lack of samples from such regions? **(Conclusions)**

  **Response:** Not really. The AFT data from several cratons such as the Kalahari, São Francisco, West African, and Tanzania cratons indicates that the Meso-Cenozoic deformation predominantly occurred along their borders and near rift zones (section 6.3, Fig. 7, and associated references). In contrast, the more internal area of these cratons, mainly yield Paleozoic AFT ages regardless of apatite chemistry, and exhibit significantly greater thermal stability, particularly when situated away from rift zones.

- **Minor Comment 35:** In this and subsequent plots, please explain this Figure. I have no knowledge of what is plotted here. **(Supplementary file)**

  **Response:** See the response to the Major Comment 8 from the "Comments from Reviewer 1".

- **Minor Comment 36:** No plots **(Supplementary file)**

  **Response:** Following the reviewer suggestion, we have reconsidered and decided to include the plots for samples with fewer than 50 measured tracks to the S2 Supplement.

- **Minor Comment 37:** No plots **(Supplementary file)**

  **Response:** Following the reviewer suggestion, we have reconsidered and decided to include the plots for samples with fewer than 50 measured tracks to the S2 Supplement.

- **Minor Comment 38:** No plots **(Supplementary file)**

  **Response:** Following the reviewer suggestion, we have reconsidered and decided to include the plots for samples with fewer than 50 measured tracks to the S2 Supplement.

- **Minor Comment 39:** No plots **(Supplementary file)**

  **Response:** Following the reviewer suggestion, we have reconsidered and decided to include the plots for samples with fewer than 50 measured tracks to the S2 Supplement.

Reply to the comments on manuscript [Extensional exhumation of cratons: insights from the Early Cretaceous Rio Negro-Juruena belt (Amazonian Craton, Colombia)] by "Fonseca et al."

- **Minor Comment 40:** No plots **(Supplementary file)**

  **Response:** Following the reviewer suggestion, we have reconsidered and decided to include the plots for samples with fewer than 50 measured tracks to the S2 Supplement.

- **Minor Comment 41:** This is MTL vs AFT age, not the other way round. This seems to be an increasingly common error. **(Supplementary file)**

  **Response:** Thank you for bringing this point to our attention. We have corrected this.

- **Minor Comment 42:** The plot itself s not a boomerang plot. Only when data show the characteristic boomerang trend of a series of samples affected to varying degrees by a common event should this term be used. And then the appropriate term is "boomerang trend". Incidentally, Green (1986) did not use the term. **(Supplementary file)**

  **Response:** We have removed this terminology and the reference from the caption of the figure.

- **Minor Comment 43:** "Projected" in what sense? **(Supplementary file)**

  **Response:** C-axis projected. We have added the complete information in the figure.

- **Minor Comment 44:** As above, this is Dpar vs AFT age. But it should be the other way round, to investigate possible variation of age with Dpar. **(Supplementary file)**

  **Response:** Agree. We have changed the x and y axis.

- **Minor Comment 45:** more explanation is needed. Particuarly in relation to the plots in the right hand column. What do these show? **(Supplementary file)**

  **Response:** We thank the reviewer for this suggestion. We have added an explanation regarding the Predicted age vs. Observed age scatterplot to the figure caption.

**Comments from Chiara Amadori (Reviewer 2)**

- **General Comment:** First, I really thank the Editor for choosing me to review the manuscript by Fonseca et al. I really enjoyed reading it and I also admit I learned a lot. Please, see my comments for this manuscript are provided in two files. Here, you find a formal summary of my major comments and concerns, and a marked-up version of the manuscript sent to me for review (PDF document). Both files contain similar comments/ questions/ suggestions. Still, comments on figures and some text corrections are better explained in the PDF document, so please ensure that both are reviewed during revisions.

Reply to the comments on manuscript [Extensional exhumation of cratons: insights from the Early Cretaceous Rio Negro-Juruena belt (Amazonian Craton, Colombia)] by "Fonseca et al."

**Response:** We appreciate the time and effort that Dr. Chiara Amadori dedicated to providing feedback on our manuscript and are grateful for the insightful comments and valuable improvements to our paper. Below, we have replied point-by-point to specific comments.

- **Major Comment 1:** The manuscript objectives are well defined but sometimes I believe the reader needs some more information or better displayed. In the reviewed PDF I have noted a few times the need to add more references. In these comments, I suggest some papers that might be included in the manuscript.

  **Response:** As suggested by the reviewer, we have amended our original manuscript, adding extra references including the suggested ones.

- **Major Comment 2:** Figure 1. In general, this figure is too crowded with information. I strongly suggest to divide it in three. 1A: DEM + Geological and geographical elements. Basins are not well highlighted if shown only with isobaths. For example, the lines behind Tukutu (or Takutu?) Basins are barely visible. If you divide this figure in two you have a cleaner DEM to put geological information on. 1B: DEM at the same scale as 1A but with only geochronological provinces, perhaps shown with polygons in different colours. 1C: geological profile. Check carefully that the geology in the profile and in the map match 100%.

  **Response:** Agree. We have, accordingly, modified the figure.

- **Major Comment 3:** In the Geological Setting you have a long description based on just two references. I believe you can integrate more bibliography. Also, you always cite fig. 1 but in that figure, it is not possible to unravel all the history you are describing.

  **Response:** Thank you for pointing this out. We regret that we missed important references. We have added references in the text (L.68-90). Incorrect citations of Figure 1 have been removed from this paragraph.

- **Major Comment 4:** In the main text, you cite Fig. 3 before Fig. 2. Switch the order of the figures If you need to.

  **Response:** Thank you for bringing this point to our attention. We have changed the order of the figures.

- **Major Comment 5:** Figure 2. What is the basement you have sampled? You show stars for ZUPb but your study is actually focused on AFT samples, thus they need to be put on this figure too.

  **Response:** Agree. We have, accordingly, modified the figure.

Reply to the comments on manuscript [Extensional exhumation of cratons: insights from the Early Cretaceous Rio Negro-Juruena belt (Amazonian Craton, Colombia)] by "Fonseca et al."

- **Major Comment 6:** Figure 4. This image is really confusing to understand... Why don't you use colour shades to show the subsidence values along the basin? Also, creating a bigger block diagram including the study area would help the reader a lot.

  **Response:** See the responses to the Minor Comments 43 – 46 from the "Comments from Reviewer 2".

- **Major Comment 7:** Figure 5. I do not fully agree with that. Your Fig. 5 shows a high peak around 1 Ga in the Neogene units. However, if you do not show the values on the y-axis you cannot compare the relative contributions from different histograms, thus your sentence here cannot be supported.

  **Response:** Agree. We have updated the figure showing the y-axis.

- **Major Comment 8:** Figure 6. These pie charts need to be larger because they show the data you discuss, so they are the most important thing in the figure. This figure is essentially a paleogeography, it is important to write – at least in the caption – the geological time the map refers to.

  **Response:** Agree. We have modified the figure.

- **Major Comment 9:** The modelling chapter is an important part of this paper. You also use an uncommon and interesting approach which deserves to be better illustrated. I suggest moving some figures from the supplemental material to the main manuscript to help the reader visualize the differences among the models' results. Reading only their descriptions can be confusing.

  **Response:** We appreciate the comment and agree with it. We have add an image to the manuscript (Fig.4) with modeling results.

- **Major Comment 10:** I see your Dpar measurements are values and sometimes very low. Do you think the differences in AFT ages and lengths are only due to the differences in thermal history and may not be influenced by the nature of the apatite itself? Perhaps you should detail this a little more.

  **Response:** Indeed, the apatite chemistry (in a way represented by the $D_{par}$ values) plays a role in fission track annealing and, consequently, AFT ages. In Supplement S3, Figure S.42, we have plotted the AFT central ages against the corresponding $D_{par}$ values. Our analysis revealed no observable correlation between these parameters in our dataset. Notably, our dataset shows minimal variation in $D_{par}$ values (approximately 0.4 μm). These slight variations are unlikely to induce substantial changes in the AFT central age based on our analysis. We appreciate your observation that this aspect was not explicitly addressed in the previous version of our manuscript. In the revised version, we have included some additional sentences in the Results section discussing the significance of our $D_{par}$ values in relation to the AFT central ages (L.222-223).

Reply to the comments on manuscript [Extensional exhumation of cratons: insights from the Early Cretaceous Rio Negro-Juruena belt (Amazonian Craton, Colombia)] by "Fonseca et al."

- **Major Comment 11:** I appreciate you were careful to not stress any conclusions for T< 60 and Cenozoic geological history. However, more and more scientists agree on the fact that one thermochron. system only can lead to a partial T-t history, and the use of two dating methods is always advised. Do you think you can use data from published works?

  **Response:** Indeed, as the reviewer indicates, we took care not to stress any conclusions for T<60°C and the corresponding Cenozoic geological history because this last part of our thermal histories models is at the limits of AFT sensitivity. Thus, it is hazardous to attach too much weight to it. We fully agree that another method with lower closure temperatures, such as potentially apatite U-Th/He (AHe) dating would be useful to try and better constrain and validate the Cenozoic cooling history. However, there are no published or publically available AHe data from our study. Moreover, the usefulness of potential AHe data is far from guaranteed considering that for cratonic regions (e.g., as observed in Kohn and Gleadow, 2019) and other older basement rocks, anomalously high AHe ages (with respect to AFT) are very frequently observed, hence making them not straightforward to incorporate in thermal history modelling.

- **Major Comment 12:** Also, you report ages from 97 to 177 Ma, but you are confident in describing the area with a homogeneous rock cooling history. I guess you need to explain this a little deeper.

  **Response:** Please, see the response to the Major Comment 4 from the "Comments from Reviewer 1".

- **Major Comment 13:** In the discussion section, I would also mention that cooling ages may also depend on the volcanic activity at the plate margin. Many passive margins are volcanic and offshore they show very interesting - and poorly constrained - Seaward Dipping Reflectors which are the product of onshore volcanic complexes today completely eroded. This means that we constantly underestimate the geothermal gradient (and its variation) and the possible reheating cycles of the margins which involves an additional complication to the T-t paths.

  **Response:** We appreciate the reviewer's feedback but disagree somewhat with her comment. While acknowledging that volcanic activity at a plate's passive margin can indeed affect basement thermal histories, we think its impact may be limited in our specific context. The equatorial Atlantic passive margin is characterized by a narrow, transforming rift zone with limited propagation of deformation and heat toward the continent (Sapin et al., 2021). Given that our study area is situated approximately 1000 km away from this margin, we raise doubts regarding the influence of margin volcanism on the thermal histories of our samples.

- **Minor Comment 1:** 'strength', **(Introduction)**

  **Response:** Agree. It has been corrected.

Reply to the comments on manuscript [Extensional exhumation of cratons: insights from the Early Cretaceous Rio Negro-Juruena belt (Amazonian Craton, Colombia)] by "Fonseca et al."

- **Minor Comment 2:** sorry, perhaps I miss the difference here. I'd say orogens and volvanic arcs. **(Introduction)**

  **Response:** Thank you for your comment. We have decided to delete the "mobile belt" terminology as there has been some discussion on its use and for our setting it is not essential to use it.

- **Minor Comment 3:** all this long (in time) geological description and just two references. I believe you can integrate more bibliography here. Also, you always cite fig. 1 but in that figure, there is a lot of information already, however, in the figure, it is not possible to unravel all the history you are describing here. **(Introduction)**

  **Response:** Agree. See the response to the Major Comment 03 from the "Comments from Reviewer 2".

- **Minor Comment 4:** in general this figure is too crowded of information. I strongly suggest to divide it in two. Figure 1A: geological and geographical elements. Figure 1B: same scale as A but with only geochronological provinces **(Fig.1)**

  **Response:** Agree. We have modified the figure.

- **Minor Comment 5:** Is the arrow indicating an area outside the map? If yes, why don't you get the map larger and include the area you are actually citing? **(Fig.1)**

  **Response:** We removed the arrow since it is not necessary and Fig.7 shows the data that is not covered in Fig.1.

- **Minor Comment 6:** this dashed red lines are a little confusing because the look like faults or tectonic lineaments. I believe you can show geochronological provinces in colors on the addisional figure I suggest to add here. **(Fig.1)**

  **Response:** Agree. We have modified the figure.

- **Minor Comment 7:** this is the same symbol you have used for the Indiscriminated or inferred fracture. Change the pattern or the color of one of them. **(Fig.1)**

  **Response:** Agree. We have modified the figure.

- **Minor Comment 8:** faults of the suture zones 1 are not shown on the map **(Fig.1)**

  **Response:** We added the suture zones on map 1A.

- **Minor Comment 9:** Perhaps you want to put numbers along the transect on the map A **(Fig.1)**

Reply to the comments on manuscript [Extensional exhumation of cratons: insights from the Early Cretaceous Rio Negro-Juruena belt (Amazonian Craton, Colombia)] by "Fonseca et al."

**Response:** We added the numbering of suture zones on the map 1A.

- **Minor Comment 10:** explain to the readers what you mean with geochronological domains. **(Introduction)**

**Response:** Agree. We have added the definition of geochronological domains( L.68-70).

- **Minor Comment 11:** is it the age of the dykes in fig. 1?

**Response:** We apologize for any confusion. In Figure 1, the majority of the dikes are from the Mesozoic, while some remain of unknown age. To prevent misunderstandings and denote their ages, we have removed the dikes with unknown ages and updated the map key to indicate the ages of the Mesozoic dikes. For your reference, we have included the dating of the dikes in the Guiana Shield below.

[Figure]

Main dykes, sills and elliptical mafic bodies within the Guiana Shield. The dashed lines indicate country borders, while the dark line marks the approximate limit of the shield. Blue colour — Paleoproterozoic–Avanavero sills and dykes (Guaniamo, Cipó, among others); dark green colour — Mesoproterozoic–anorthosite/gabbro and dykes (Repartimento, Käyser); light green colour (1.88–1.17 Ga) — mafic–ultramafic bodies (De Goeje, Estrutura, Uraricaá and Tapuruquara); pink colour—Neoproterozoic-dykes (Tampok); red colour — Mesozoic-dykes (Taiano, Apoteri, Apatoe, Cassiporé, Penatecaua and Uaraná); and black colour — unknown age. Figure from Reis et al., 2013.

- **Minor Comment 12:** in Fig. 1 you wrote different ages: 1.9-1.8 **(Geological Setting)**

**Response:** We regret the inconsistency between our original statement and the figure's information. To rectify this, we have updated the statement to ensure agreement between the information provided in the figure and the accompanying text. Additionally, in Figure 1, the Rondonian-San Ignacio (ca. 1.5-1.3 Ga) and Putumayo (1.45–0.98 Ga) belts are entirely covered by the Phanerozoic basins, thus not depicted on the map. The map key in the figure has been updated to clarify that the geochronological units mentioned refer specifically to the exposed rock provinces.

Reply to the comments on manuscript [Extensional exhumation of cratons: insights from the Early Cretaceous Rio Negro-Juruena belt (Amazonian Craton, Colombia)] by "Fonseca et al."

- **Minor Comment 13:** not represented in Fig. 1 **(Geological Setting)**

  **Response:** See the response to the Minor Comment 12 from the "Comments from Reviewer 2".

- **Minor Comment 14:** Putumayo in Fig. 1 is on a basin and here had ages and described as a belt. I'm confused. **(Geological Setting)**

  **Response:** See the response to the Minor Comment 12 from the "Comments from Reviewer 2".

- **Minor Comment 15:** and fig. 2? **(Geological Setting)**

  **Response:** Agree. See the response to the Major Comment 4 from the "Comments from Reviewer 2".

- **Minor Comment 16:** sometimes you write 1.9-2 Ga or 1.8 and now 1.85. Is it always the same phase? if yes, use same age interval **(Geological Setting)**

  **Response:** See the response to the Minor Comment 12 from the "Comments from Reviewer 2".

- **Minor Comment 17:** I think you have to set an order here. Either you decide to order the references sequencially per year (younger to older) or alphabetically (A to Z) is fine, but maintain the same order over the manusctipt. **(Geological Setting)**

  **Response:** Thank you for bringing this point to our attention. We have set the reference order by time (younger to older).

- **Minor Comment 18:** In the text, you cite fig. 3 before fig. 2. switch the order of the figures If you need to. **(Geological Setting)**

  **Response:** Agree. See the response to the Major Comment 4 from the "Comments from Reviewer 2".

- **Minor Comment 19:** What is the basement you have sampled? you show stars for ZUPb but your study is actually focused on AFT samples, thus they need to be put on this figure too. **(Fig.2)**

  **Response:** Agree. We have, accordingly, modified the figure.

- **Minor Comment 20:** in fig 1 you wrote TUKUTU, correct the typo **(Geological Setting)**

  **Response:** Thank you for pointing this out. We have corrected this.

Reply to the comments on manuscript [Extensional exhumation of cratons: insights from the Early Cretaceous Rio Negro-Juruena belt (Amazonian Craton, Colombia)] by "Fonseca et al."

- **Minor Comment 21:** which ones? please explain it better. **(Geological Setting)**

  **Response:** Thank you for your comment. We have added information over the lithologies of these deposits (L.159-160).

- **Minor Comment 22:** show them in fig 2 too **(Samples and methods)**

  **Response:** Agree. See the response to the Minor Comment 19 from the "Comments from Reviewer 2".

- **Minor Comment 23:** sample ID **(Tab.1)**

  **Response:** Agree. We have added the "ID".

- **Minor Comment 24:** in the caption you also have to write that the data are displayed on map in fig. 3. **(Tab.1)**

  **Response:** Agree. We have included this information to the caption of the figure.

- **Minor Comment 25:** You do not need an acronym if you only use it here. There is no need to mention the OMWZ later on in the text, thus you do not need to create an acronym from the overall weighted mean zeta. **(Samples and methods)**

  **Response:** Agree. We have deleted the acronym.

- **Minor Comment 26:** What this brown color represent? What does the star represent? **(Fig.3)**

  **Response:** The brown color represents the Ventuari-Tapajos province and the star is the location of Neblina peak. To make it clear we have added this information in the map key.

- **Minor Comment 27:** what do the dark and light grey colors represent?st ref figure 1 once. **(Fig.3)**

  **Response:** The dark and light grey colors represent the Rio Negro-Juruena province and Phanerozoic covers, respectively. To make it clear we have added this information in the map key.

- **Minor Comment 28:** another weird wavy "fault"...is it possible that you colored in black another pattern? **(Fig.3)**

  **Response:** We do not know the reason for this "wavy" contour in the map. The original map is available at https://rigeo.cprm.gov.br/jspui/handle/doc/22532 and we attached below a print screen of the area and the key for the structures. It is really difficult to

distinguish between the features. Our interpretation leans towards labeling it as an 'Inferred normal fault' due to the presence of Proterozoic rift deposits. Yet, it is ambiguous whether it is a fault or a lineament. To prevent potential misunderstanding, we have updated these lines to be categorized as lineaments. This adjustment aims to accommodate both possibilities, recognizing that even if it is a fault, it can also be considered a lineament.

[Figure]

[Figure]

- **Minor Comment 29:** If these black dashed lines are normal faults (as the legend says), the have a very weird attitude! they curve in a way that I do not understand the overall structure. **(Fig.3)**

  **Response:** See the response to the Minor Comment 28 from the "Comments from Reviewer 2".

- **Minor Comment 30:** space missing **(Results)**

  **Response:** Thank you for pointing this out. We have added the missing space.

- **Minor Comment 31:** The modelling chapter is an important part of this paper. You also use an uncommon and interesting approach which deserves to be better illustrated. I suggest moving some figures from the supplemental material to the main manuscript and helping the reader to visualize the differences among the models. **(Thermal History Modelling)**

  **Response:** See the response to the Major Comment 9 from the "Comments from Reviewer 2".

- **Minor Comment 32:** ten to the 5, otherwise it is one hundred and five. **(Thermal History Modelling)**

  **Response:** Agree. We have corrected the typo mistake.

Reply to the comments on manuscript [Extensional exhumation of cratons: insights from the Early Cretaceous Rio Negro-Juruena belt (Amazonian Craton, Colombia)] by "Fonseca et al."

- **Minor Comment 33:** I see your Dpar measurements show low values, do you think the differences in AFT ages and lengths are only due to the differences in thermal history and may not be influenced by nature of apatite itself? **(Thermal History Modelling)**

  **Response:** See the response to the Major Comment 10 from the "Comments from Reviewer 2".

- **Minor Comment 34:** this column is not fundamental. You already show spontaneous and induces track densities. If a reader needs the ratio, can self-calculate it. **(Tab.2)**

  **Response:** Agree. We have deleted this column.

- **Minor Comment 35:** unit (micron) is missin. **(Tab.2)**

  **Response:** Thank you for pointing this out. We have added the unit to the $D_{par}$ column.

- **Minor Comment 36:** in the supplemental material I see your Dpar has large variuabiliy, perhaps you should show a sigma or SD too. At least, you have to clear in the caption that this is your mean Dpar. **(Tab.2)**

  **Response:** Agree. We have mentioned that the value is the mean $D_{par}$.

- **Minor Comment 37:** rotate 90 deg this table to landscape view. Data and text will be better readable **(Tab.2)**

  **Response:** We think that the updated version of the table is readable on the horizontal.

- **Minor Comment 38:** generally the dosimeter is also specified in table captions **(Tab.2)**

  **Response:** Agree. We have added to the table caption.

- **Minor Comment 39:** You need to define the Acceptance rates: what is it? why is it important? **(Thermal History Modelling)**

  **Response:** Thank you for your suggestion. We have added the importance of evaluating acceptance rates to the Supplement S4 and add a sentence in the manuscript to refer to this explanation (L.235).

- **Minor Comment 40:** during Cenozoiz. I'd remove the brakets. **(Discussion)**

  **Response:** Agree. We have modified the sentence.

- **Minor Comment 41:** you show ages from 97 to 177 Ma, are you sure this can be describes as homogeneous rock cooling history???? **(Discussion)**

Reply to the comments on manuscript [Extensional exhumation of cratons: insights from the Early Cretaceous Rio Negro-Juruena belt (Amazonian Craton, Colombia)] by "Fonseca et al."

**Response:** Please, see the response to the Major Comment 4 from the "Comments from Reviewer 1".

- **Minor Comment 42:** the most recent paper on CAMP dating is Davies, J. H. F. L. et al. End-Triassic mass extinction started by intrusive CAMP activity. Nat. Commun. 8, 15596 doi: 10.1038/ncomms15596 (2017). **(Discussion)**

  **Response:** We regret that we did not cite this reference. We added it (L.337) to the text.

- **Minor Comment 43:** you already use A B C in this figure, why don't you use roman numbers, like I, II, III or actual numbers? **(Fig. 4)**

  **Response:** Agree. We have modified the figure.

- **Minor Comment 44:** here a reference is missing. You have a reference for all other info, it's needed here too. **(Fig. 4)**

  **Response:** Agree. We have modified the figure.

- **Minor Comment 45:** this image is really confusing to understand.... Why don't you use color shades to show the subsidence values along the basin? Also creating a bigger drawing including the sudy area would help the reader a lot. **(Fig. 4)**

  **Response:** While the first part of the comment regarding the subsidence it not clear to us, we have now referred to the study area in the drawing, although the diagram remains a very schematical representation of our interpretation. We hope this modification helps improve the clarity and aids the readers to better understand the context.

- **Minor Comment 46:** In the figure you do not need to put A B and C because you have enough space to write the name of the basins in figure c. **(Fig. 4)**

  **Response:** We have tried to write the names in the Figure c but was not enough space to "Llanos Orientales", so we kept the numbering.

- **Minor Comment 47:** order the references by name or year **(Discussion)**

  **Response:** Agree. See the response to the Minor Comment 17 from the "Comments from Reviewer 2".

- **Minor Comment 48:** one reference is not enough, check the latest ones too:
  K.E. Murray, P.W. Reiners, S.N. Thomson; Rapid Pliocene–Pleistocene erosion of the central Colorado Plateau documented by apatite thermochronology from the Henry Mountains. Geology 2016;; 44 (6): 483–486. doi: https://doi.org/10.1130/G37733.1

Karl E. Karlstrom, Justin Wilgus, Jacob O. Thacker, Brandon Schmandt, David Coblentz, Micael Albonico. Tectonics of the Colorado Plateau and Its Margins. Annual Review of Earth and Planetary Sciences 2022 50:1, 295-322

Christian Rønnevik, Anna K. Ksienzyk, Haakon Fossen, Joachim Jacobs; Thermal evolution and exhumation history of the Uncompahgre Plateau (northeastern Colorado Plateau), based on apatite fission track and (U-Th)-He thermochronology and zircon U-Pb dating. Geosphere 2017;; 13 (2): 518–537. doi: https://doi.org/10.1130/GES01415.1

Heitmann EO, Hyland EG, Schoettle-Greene P, Brigham CAP and Huntington KW (2021) Rise of the Colorado Plateau: A Synthesis of Paleoelevation Constraints From the Region and a Path Forward Using Temperature-Based Elevation Proxies. Front. Earth Sci. 9:648605. doi: 10.3389/feart.2021.648605 **(Discussion)**

**Response:** Thank you for the suggested references. They were checked and added accordingly.

- **Minor Comment 49:** Cetiting only this paper is not fully correct. This topic is getting hotter in the scientific community thus several good papers are not out since 2010, for example:

Ding, L., Kapp, P., Cai, F. et al. Timing and mechanisms of Tibetan Plateau uplift. Nat Rev Earth Environ 3, 652–667 (2022). https://doi.org/10.1038/s43017-022-00318-4

Yang Y, Nie J, Miao Y, Wan S and Jonell TN (2022), Editorial: Tibetan Plateau uplift and environmental impacts: New progress and perspectives. Front. Earth Sci. 10:1020354. doi: 10.3389/feart.2022.1020354

Furlong KP, Kirby E, Creason CG, Kamp PJJ, Xu G, Danišˇ ík M, Shi X and Hodges KV (2021) Exploiting Thermochronology to Quantify Exhumation Histories and Patterns of Uplift Along the Margins of Tibet. Front. Earth Sci. 9:688374. doi: 10.3389/feart.2021.688374

Kui Tong, Zhiwu Li, Lidong Zhu, Ganqing Xu, Yuxiu Zhang, Peter J.J. Kamp, Gang Tao, Wenguang Yang, Jinxi Li, Zijian Wang, Xun Jiang, Haosheng Zhang, Thermochronology constraints on the Cretaceous-Cenozoic thermo-tectonic evolution in the Gaize region, central-western Tibetan Plateau: Implications for the westward extension of the proto-Tibetan Plateau, Journal of Asian Earth Sciences, Volume 240, 2022, 105419, ISSN 1367-9120, https://doi.org/10.1016/j.jseaes.2022.105419. **(Discussion)**

**Response:** Thank you for the suggested references. They were checked and added accordingly.

- **Minor Comment 50:** in both Z U-Pb collection the y-axis is not explained. Frequency values should be shown as well. **(Discussion)**

**Response:** See the response to the Major Comment 7 from the "Comments from Reviewer 2".

- **Minor Comment 51:** I see your AFT interpretation leads to this conclusion. However, you were careful to not stress T< 60 and Cenozoic geological history. One system only can

be very helpful but if you want to detail the younger and cooler paths I believe you must introduce another appropriate thermochronometer. **(Discussion)**

**Response:** See the response to the Major Comment 11 from the "Comments from Reviewer 2".

- **Minor Comment 52:** I agree. Once you write this, a general reader can be suspicious about all the rest. **(Discussion)**

**Response:** In this part of the text we highlight that heating/cooling events at lower temperatures ca. <60°C are not well constrained by our models. According to our data, our samples experienced these lower temperatures since the early Cenozoic, implying lower erosion and exhumation rates of the craton during the Cenozoic. The use of other thermochronometers could potentially better constrain the events during that time. We regret that our original statement did not make this clear. We have now added a sentence clarifying this point (L.406-408).

- **Minor Comment 53:** I do not fully agree with that. Your fig. 5 shows a high peak around 1 Ga in the Neogene units. However, if you do not show the values on the y-axis you cannot compare all the histograms thus your sentence here cannot be supported. **(Discussion)**

**Response:** See the response to the Major Comment 7 from the "Comments from Reviewer 2".

- **Minor Comment 54:** I understand you compare AFT ages because it's the same thermochronometer used across the whole area. However., I'm suere there are other data available that you can show to help the reader to understan the overall evolution. Is there no other AHe oer ZHe dataset available? **(Discussion)**

**Response:** Our interpretation indeed primarily relies on AFT ages, as the available information from other thermochronometers, at present, seems more suitable for localized, regional-scale analysis, and is very limited at best.

- **Minor Comment 55:** this pinkish color assigned to Mesozoic-Cenozoic Orogen looks exactly the same color of Triassic in the age column on the bottom right. If you run out of colors, use patterns. **(Fig.6)**

**Response:** Agree. We have modified the figure.

- **Minor Comment 56:** Please, get these pie charts larger. They show the data you discuss so they are the most important thing to show in the figure. **(Fig.6)**

**Response:** See the response to the Major Comment 08 from the "Comments from Reviewer 2".

Reply to the comments on manuscript [Extensional exhumation of cratons: insights from the Early Cretaceous Rio Negro-Juruena belt (Amazonian Craton, Colombia)] by "Fonseca et al."

- **Minor Comment 57:** Did these work provide AFT only? **(Discussion)**

  **Response:** See the response to the Minor Comment 54 from the "Comments from Reviewer 2".

- **Minor Comment 58:** to the power 2 **(Discussion)**

  **Response:** Agree. We have corrected the typo mistake.

- **Minor Comment 59:** I would also say that it depends on the volcanic activity at the plate margin. Many passive margins are volcanic and offshore they show very interesting - and poorly constrained - Seaward Dipping Reflectors which are the product of volcanic complexes completely eroded. This means that we constantly underestimate the geothermal gradient and the possible reheating effect of the margins which involves an additional complication to the T-t paths. **(Discussion)**

  **Response:** See the response to the Major Comment 13 from the "Comments from Reviewer 2".

- **Minor Comment 60:** Deleting typo **(Discussion)**

  **Response:** Agree. We have deleted the comma.
* * *
  ***Supplementary References***

  Reis, N. J., Teixeira, W., Hamilton, M. A., Bispo-Santos, F., Almeida, M. E., & D'Agrella-Filho, M. S. (2013). Avanavero mafic magmatism, a late Paleoproterozoic LIP in the Guiana Shield, Amazonian Craton: U–Pb ID-TIMS baddeleyite, geochemical and paleomagnetic evidence. Lithos, 174, 175-195.

  Sapin, F., Ringenbach, J. C., & Clerc, C. (2021). Rifted margins classification and forcing parameters. Scientific Reports, 11(1), 1–17. https://doi.org/10.1038/s41598-021-87648-3